# VinePPO: Refining Credit Assignment in RL Training of LLMs

Amirhossein Kazemnejad [* 1]  Milad Aghajohari [* 1]  Eva Portelance [1 6]  Alessandro Sordoni [1 2]  Siva Reddy [1 3 4]
Aaron Courville [† 1 4 5]  Nicolas Le Roux [† 1 4]

## Abstract

Large language models (LLMs) are increasingly applied to complex reasoning tasks that require executing several complex steps before receiving any reward. Properly assigning credit to these steps is essential for enhancing model performance. Proximal Policy Optimization (PPO), a common reinforcement learning (RL) algorithm used for LLM finetuning, employs value networks to tackle credit assignment. However, recent approaches achieve strong results without it, raising questions about the efficacy of value networks in practice. In this work, we systematically evaluate the efficacy of value networks and reveal their significant shortcomings in reasoning-heavy LLM tasks, showing that they often produce poor estimate of expected return and barely outperform a random baseline when comparing alternative steps. This motivates our key question: Can improved credit assignment enhance RL training for LLMs? To address this, we propose VinePPO, a straightforward approach that leverages the flexibility of language environments to compute unbiased Monte Carlo-based estimates. Our method consistently outperforms PPO and other baselines across MATH and GSM8K datasets in less wall-clock time (up to 3.0x). Crucially, it achieves higher test accuracy for a given training accuracy, capturing more generalization signal per sample. These results emphasize the importance of accurate credit assignment in RL training of LLM.

Code available at https://github.com/McGill-NLP/VinePPO

---

[*]Equal contribution [†]Equal advising [1]Mila [2]Microsoft Research [3]McGill University [4]Canada CIFAR AI Chair [5]Université de Montréal [6]HEC Montréal. Correspondence to: Amirhossein Kazemnejad <amirhossein.kazemnejad@mila.quebec>, Milad Aghajohari <aghajohm@mila.quebec>.

*Proceedings of the $42^{nd}$ International Conference on Machine Learning*, Vancouver, Canada. PMLR 267, 2025. Copyright 2025 by the author(s).

## 1. Introduction

Reinforcement learning (RL) has become instrumental in training large language models (LLMs) to solve complex reasoning tasks such as mathematical problem solving (DeepSeek-AI et al., 2025), web navigation (Putta et al., 2024), or code generation (OpenAI, 2024). In these settings, LLMs often engage in extended reasoning steps, executing multiple actions to arrive at a solution. However, not all steps are equally impactful—some contribute significantly, while others are irrelevant or detrimental. For example, in Figure 1.a, only step $s_2$ provides a key insight. Indeed, most reasoning steps generated by a model do not affect the chance of it solving the problem (Figure 1.b). Identifying the contribution of each action is crucial for improving model performance. However, this is inherently difficult due to the significant delay between actions and their eventual effect. This issue, known as the *Credit Assignment (CA)* problem, is a core challenge in RL (Sutton & Barto, 1998).

Proximal Policy Optimization (PPO; Schulman et al. 2017) addresses credit assignment through a value network (or critic), a mechanism retained in its application to RL-based finetuning of LLMs (Ouyang et al., 2022). This network, typically a separate model initialized from a pretrained checkpoint, is trained during PPO finetuning to estimate the expected cumulative rewards (or value) of an intermediate action. In Figure 1.b, an ideal value network would assign high value to step $s_2$ and subsequent steps, where the model has a high chance of successfully solving the problem. PPO uses these value estimates to measure the *advantage* of each action and update the model accordingly.

However, recent approaches such as DPO (Rafailov et al., 2023) or GRPO (Shao et al., 2024) simplify PPO's design by discarding fine-grained credit assignment and treating all tokens equally. Despite such simplifications, they often demonstrate strong performance (Xu et al., 2024; Chang et al., 2023). This challenges classic RL principles, where accurate CA is considered critical for optimal performance (Sutton & Barto, 1998; Greensmith et al., 2001), especially in tasks with delayed rewards. In this work, we address this apparent discrepancy by showing that PPO's credit assignment mechanism, the value network, performs poorly in practice. Our systematic evaluation (Section 7) on tasks

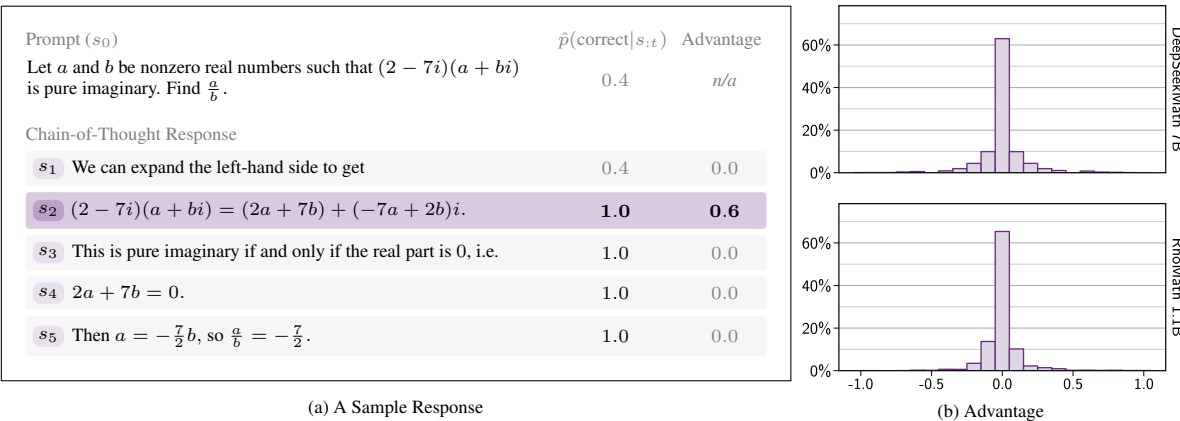

Figure 1: **(Left)** A response generated by the model. The notation $\hat{p}(\text{correct}|s_{:t})$ represents the estimated probability of successfully solving the problem at step $t$. Here, only step $s_2$ is critical; after this, the model always completes the solution correctly. **(Right)** The distribution of advantages, defined as $\hat{p}(\text{correct}|s_{:t+1}) - \hat{p}(\text{correct}|s_{:t})$, collected over a subset of MATH dataset (Hendrycks et al., 2021). Most steps show little or no advantage over the preceding step.

requiring chain-of-thought reasoning reveals that the value network often provides imprecise estimates and fails to differentiate between promising and unproductive steps, which could explain why simplified approaches achieve comparable results without explicit fine-grained CA.

These findings motivate a central question: If we improve credit assignment in PPO rather than discarding it, *how much can we enhance the RL training of LLMs?* To explore this, we propose VinePPO (Figure 2), which computes *unbiased* value estimates of the intermediate states with Monte Carlo (MC) estimation instead of employing value networks. Our key insight is that language-based environments allow us to reset directly to any intermediate state simply by re-feeding the partial context, enabling efficient MC rollouts without the massive overhead usually seen in generic RL environments. VinePPO preserves PPO's overall framework but addresses the CA challenge fundamentally.

We empirically evaluate the effectiveness and computational efficiency of MC value estimation in VinePPO. Across multiple mathematical reasoning tasks, VinePPO consistently outperforms PPO and other credit assignment-free baselines. While its per-iteration runtime is generally slower due to MC sampling, VinePPO surpasses the peak performance of baselines with fewer gradient steps and ultimately less wall-clock time. Importantly, VinePPO achieves higher test accuracy for a given training accuracy, capturing more generalization signal per fitted training sample. This is critical, as genuinely challenging verifiable reasoning tasks are scarce. These results underscore the importance of CA in RL-training of LLMs and highlight VinePPO as a straightforward alternative to value network-based approaches.

Our contributions are summarized as follows:

- We analyze PPO's value network in reasoning tasks and find it often misestimates intermediate values, barely outperforming a random chance in ranking candidate steps.

- We propose VinePPO, leveraging the flexibility of language environments to compute unbiased, MC-based value estimates without relying on a separate critic.

- We empirically highlights the benefits of refined CA. VinePPO achieves the peak performance of baselines with less wall-clock time (up to 3.0x), better KL-divergence trade-off while exhibiting better generalization slope.

## 2. Related Work

**Credit Assignment in Post-Training of LLM**   PPO, as applied in RL from Human Feedback (RLHF, Ouyang et al. 2022), pioneered RL finetuning of LLMs. However, its computational overhead and hyperparameter sensitivity led to the development of simpler alternatives. RL-free methods such as DPO (Rafailov et al., 2023) operate in a bandit setting, treating the entire response as a single action. Similarly, rejection sampling methods like RestEM (Singh et al., 2024) finetune on full high-reward responses. RLOO (Ahmadian et al., 2024) and GRPO (Shao et al., 2024) abandon PPO's value network, instead using average reward from multiple samples as a policy gradient baseline. Recent work has emphasized finer credit assignment, with Hwang et al. (2024) and Setlur et al. (2024) introducing MC-based methods to detect key errors in reasoning chains for use as ad-hoc mechanisms in DPO. Our work, by contrast, fully embraces the RL training, with the target of fixing CA in principle. Parallel efforts have also focused on building better verifiers and reward models for per-step feedback, with recent attempts to automate their data collection using MC rollouts

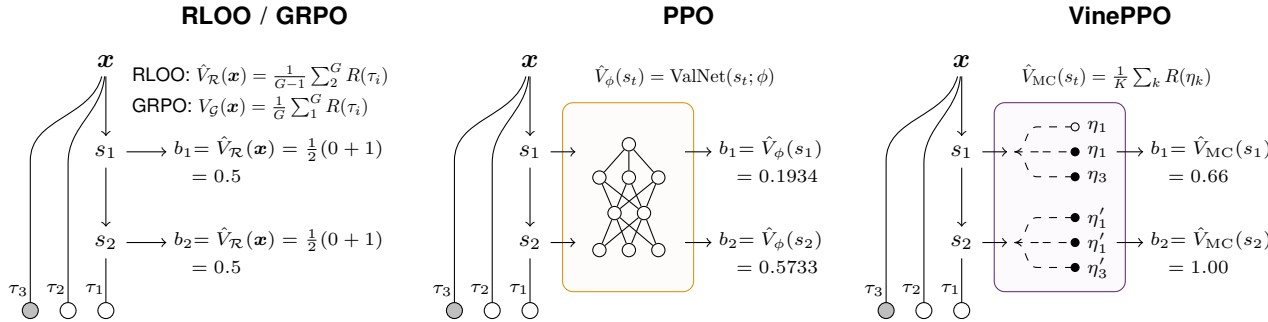

Figure 2: Comparison of credit assignment mechanisms applied on training trajectories $\tau_i$'s, depicted for states $s_1$ and $s_2$. **(Left)** RLOO and GRPO both treat all intermediate states equally and use the average return of trajectory group $\tau_i \sim \pi(\cdot|x)$ for the policy-gradient baselines $b_1$ and $b_2$. GRPO additionally normalize these returns to have a unit variance. In the case of RLOO, the computed baseline could be viewed as MC estimate of value but solely for the initial state. **(Middle)** PPO trains a separate model to predict values for each state $s_t$. **(Right)** VinePPO generate auxiliary rollouts $\eta_k \sim \pi(\cdot|s_t)$ to obtain MC estimate of state $s_t$'s value. Note that $\eta_k$'s are only used for value estimation—not to update the policy directly.

(Ma et al., 2023; Uesato et al., 2022; Luo et al., 2024; Wang et al., 2024). Our method is orthogonal to these methods, operating within PPO-based training to optimize a *given* reward, instead of designing new ones.

**Value Estimation in RL and Monte Carlo Tree Search (MCTS)** Deep RL algorithms are typically categorized into value-based and policy-based methods. Policy-based methods like PPO usually employ critic networks for value prediction. An exception is the *"Vine"* variant of TRPO (Schulman et al., 2015), which uses MC samples for state value estimation. The authors, however, note that the Vine variant is limited to environments that allow intermediate state resets, rare in typical RL settings[1]. However, language generation – when formulated as RL environment – enables such intermediate reset capabilities. In domains with similar reset capabilities, such as Go and Chess, MC-based methods like AlphaGo (Silver et al., 2016) and AlphaZero (Silver et al., 2017) have emerged. AlphaGo's architecture includes a policy, trained using expert moves and self-play, and a value network that predicts game outcomes. At inference, it employs tree search guided by MC rollouts and value network to select optimal moves. AlphaZero advances this approach by distilling MCTS outcomes into the policy. Recent works have adapted AlphaZero's principles to LLMs, employing similar search techniques for inference and trajectory distillation (Xie et al., 2024; Chen et al., 2024; Wan et al., 2024; Zhang et al., 2024; Hao et al., 2023). While this is a promising direction, our method is not an MCTS approach; it uses MC samples solely for value estimation during PPO *training* to improve credit assignment.

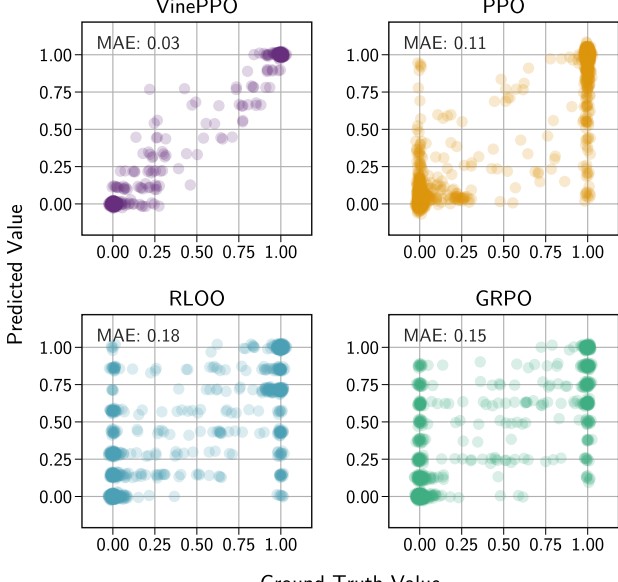

Figure 3: Distribution of predicted values for each state vs. ground truth (see Section 7 for details) for DeepSeekMath 7B on the MATH dataset, highlighting the nature of errors: PPO exhibits biased value predictions, whereas VinePPO remains unbiased. Note that RLOO/GRPO do not predict values; we plot their computed baselines against the ground truth value solely for demonstration.

## 3. Background

We focus on the RL tuning phase, following Ouyang et al. (2022); Shao et al. (2024). In this section, we provide an overview of actor-critic finetuning as implemented in the standard PPO framework.

---

[1]This is reflected in the design of Gym (Towers et al., 2024), which only allows resets to the initial state.

**RL Finetuning** In this setup, the policy $\pi_\theta$ represents a language model that generates a response $\boldsymbol{y} = [y_0, \ldots, y_{T-1}]$ autoregressively given an input $\boldsymbol{x} = [x_0, \ldots, x_{M-1}]$. The goal of RL finetuning is to maximize the expected undiscounted ($\gamma = 1$) finite-horizon return, while incorporating a KL-divergence constraint to regularize the policy and prevent it from deviating too far from a reference policy $\pi_{\text{ref}}$ (typically the initial supervised finetuned, SFT, model). The objective can be written as:

$$J(\theta) = \mathbb{E}_{\boldsymbol{x} \sim \mathcal{D}, \boldsymbol{y} \sim \pi(\cdot|\boldsymbol{x})} \left[ \mathcal{R}(\boldsymbol{x}; \boldsymbol{y}) \right] - \beta \operatorname{KL}[\pi_\theta \| \pi_{\text{ref}}], \quad (1)$$

where $\mathcal{D}$ is the dataset of prompts, $\mathcal{R}(\boldsymbol{x}; \boldsymbol{y})$ is the sequence-level reward function, and $\beta$ controls the strength of the KL penalty. Note that the policy $\pi_\theta$ is initialized from $\pi_{\text{ref}}$.

**Language Environment as an MDP** Language generation is typically modeled as a token-level Markov Decision Process (MDP) in an actor-critic setting, where each response $\boldsymbol{y}$ is an episode. The state at time step $t$, $s_t \in \mathcal{S}$, is the concatenation of the input prompt and the tokens generated up to that point: $s_t = \boldsymbol{x}; \boldsymbol{y}_{<t} = [x_0, \ldots, x_{M-1}, y_0, \ldots, y_{t-1}]$. At each time step, the action $a_t$ corresponds to generating the next token $y_t$ from fixed vocabulary. The process begins with the initial state $s_0 = \boldsymbol{x}$, and after each action, the environment transitions to the next state, $s_{t+1} = s_t; [a_t]$, by appending the action $a_t$ to the current state $s_t$. In this case, since states are always constructed by concatenating tokens, the environment dynamics are known and the transition function is *deterministic*, i.e., $P(s_{t+1}|s_t, a_t) = 1$. During the generation process, the reward $r_t$ is set to zero for all intermediate actions $a_t$'s, with the sequence-level reward $\mathcal{R}(\boldsymbol{x}; \boldsymbol{y})$ only applied at the final step when the model stops generating. A trajectory $\tau = (s_0, a_0, s_1, a_1, \ldots)$ is therefore a sequence of state-action pairs, starting from the input prompt until the terminal state. Finally, we define the cumulative return of a trajectory $\tau$ as $R(\tau) = \sum_{t=0}^{T-1} r_t = r_{T-1} = \mathcal{R}(\boldsymbol{x}; \boldsymbol{y})$.

**Policy Gradient** Given this MDP formulation, policy gradient methods like PPO maximize Equation 1 by repeatedly sampling trajectories and taking a step in the direction of the gradient $\boldsymbol{g}_{\text{pg}} \coloneqq \nabla_\theta J(\theta)$ to update the parameters. Policy gradient $\boldsymbol{g}_{\text{pg}}$ takes the following form:

$$\boldsymbol{g}_{\text{pg}} = \mathbb{E}_{\tau \sim \pi_\theta} \left[ \sum_{t=0}^{T-1} \nabla_\theta \log \pi_\theta(a_t|s_t) A(s_t, a_t) \right], \quad (2)$$

where $s_t = \boldsymbol{x}; \boldsymbol{y}_{<t}$, $a_t = y_t$, and $A(s_t, a_t)$ is the *advantage* function. If $A(s_t, a_t) > 0$, $\boldsymbol{g}_{\text{pg}}$ will increase the probability of action $a_t$ in state $s_t$, and decrease it when $A(s_t, a_t) < 0$. Intuitively, the advantage function quantifies how much better action $a_t$ is compared to average actions taken in state

$s_t$ under the policy. Formally, it is defined as:

$$\begin{aligned} A(s_t, a_t) &= Q(s_t, a_t) - V(s_t) \\ &= r_t + \gamma V(s_{t+1}) - V(s_t). \end{aligned} \quad (3)$$

where $Q(s_t, a_t)$ is the state-action value and $V(s_t)$ is the per-state value function[2]. The value function, $V(s_t) : \mathcal{S} \to \mathbb{R}$, offers a long-term assessment of how desirable a particular state is under the current policy. Formally, it represents the expected cumulative reward obtained from starting in state $s_t$ and following the policy thereafter[3]: $V(s_t) = \mathbb{E}_{\tau \sim \pi_\theta} \left[ R(\tau) \mid s_0 = s_t \right]$. PPO uses the same advantage-weighted policy gradient as in Equation 2, but constrains policy updates through clipping to ensure stable training. For full details, see Appendix A.

**Estimating Advantage via Value Networks** In practice, the advantage $A(s_t, a_t)$ is not known beforehand and is typically estimated by first using a value network $\hat{V}_\phi$ to approximate the *true value function* $V(s_t)$, then substituting the learned values into Equation 3 or alternative methods like GAE (Schulman et al., 2016). The value network is parameterized by $\phi$ and trained alongside the policy network $\pi_\theta$. The training objective for the value network minimizes the mean squared error between the predicted value and the empirical return:

$$\mathcal{L}_V(\phi) = \mathbb{E}_{\tau \sim \pi_\theta} \left[ \frac{1}{T} \sum_t \frac{1}{2} (\hat{V}_\phi(s_t) - G_t)^2 \right], \quad (4)$$

where $G_t = \sum_{t'=t}^{T-1} r_{t'}$ is the empirical return from state $s_t$. PPO uses the same objective for $\hat{V}_\phi$ but applies clipping for training stability (see Appendix A.1 for details). In RL-tuning of LLMs, the value network is often initialized using the initial SFT policy $\pi_{\text{ref}}$ (or the reward model when available), with the language modeling head swapped out for a scalar head to predict values (Zheng et al., 2023). This setup leverages the prior knowledge of the pretrained model.

## 4. Accurate Credit Assignment with VinePPO

As outlined in Section 3, a step in the PPO gradient update aims to increase the probability of better-than-average actions while decreasing the probability of those that perform worse—a process quantified by the advantage $A(s_t, a_t)$. However, the true advantage is generally unknown and must be estimated, typically by substituting estimates from a value network into Equation 3. As we will elaborate in Section 7, value networks are often inaccurate and result in biased value computation. Fortunately, the language environment as an MDP (Section 3) offers a useful property that

---

[2]Such derivation is possible as the language environment is deterministic.

[3]We drop the dependency on $\pi_\theta$ for brevity.

allows for unbiased estimation of $V(s_t)$. Since states are simply concatenated tokens, we can prompt the language model $\pi_\theta$ to generate continuations from any intermediate state. This flexibility allows us to explore alternative future paths from arbitrary points in a generation.

Specifically, computing the advantage requires access to $V(s_t) = \mathbb{E}\left[R(\tau) \mid s_0 = s_t\right]$. VinePPO obtain an MC estimation of this expectation by randomly sampling continuations and averaging their returns. That is, for each state $s_t$ in a training trajectory $\tau$, we utilize the resetting property and re-feed the partial context corresponding to $s_t$ to the current policy to sample $K$ auxiliary rollouts $\eta_1, \ldots, \eta_K \sim \pi_\theta(\cdot \mid s_t)$. The empirical mean of returns across these rollouts serves as the value estimate:

$$\hat{V}_{\mathrm{MC}}(s_t) = \frac{1}{K}\sum_{k=1}^{K} R(\eta_k). \qquad (5)$$

Critically, $\eta_k$'s are used exclusively for value estimation and do not contribute directly to policy gradient updates as we lack CA on them. Once the value $\hat{V}_{\mathrm{MC}}(s_t)$ is computed, we estimate the advantages of each action using Equation 3:

$$\hat{A}_{\mathrm{MC}}(s_t, a_t) = r(s_t, a_t) + \gamma\hat{V}_{\mathrm{MC}}(s_{t+1}) - \hat{V}_{\mathrm{MC}}(s_t). \quad (6)$$

For any $K \geq 1$, the policy gradient computed using the advantage estimator $\hat{A}_{\mathrm{MC}}$ is an unbiased estimate of the gradient of expected return $g_{\mathrm{pg}}$. PPO framework then uses $\hat{A}_{\mathrm{MC}}$ to update the policy on trajectory $\tau$.

Variance and computational efficiency represent core trade-offs in every Monte Carlo estimation. Here, the sampling parameter $K$ control such tradeoff— increasing $K$ reduces estimator variance at the expense of increased sampling demands. In Section 6, we rigorously characterize these properties for VinePPO.

To enhance the efficiency of $\hat{A}_{\mathrm{MC}}$, we group states within a reasoning step and compute a single advantage, which is assigned to all tokens in that step (examples in Appendix B). This trades off granularity for efficiency, allowing finer resolution with more compute, or coarser estimates with limited resources. Furthermore, modern LLM inference engines (Kwon et al., 2023; Zheng et al., 2024) enable rapid on-the-fly generation[4], making our MC-based approach computationally practical at scale.

By restricting modifications only to the advantage computation stage of PPO, our approach also isolates the effects of improved credit assignment, revealing how unbiased advantage estimation fundamentally alters policy optimization dynamics compared to value-network baselines.

---

[4]Achieving up to 5K tokens/second on a single Nvidia A100 GPU for 7B LLMs in bfloat16.

## 5. Experimental Setup

**Datasets and Pretrained LLMs**   We conduct experiments on publicly available LLMs and datasets to ensure reproducibility. We use base versions of DeepSeekMath 7B (Shao et al., 2024) and RhoMath 1.1B (Lin et al., 2024) which are pretrained on mathematical and natural language corpora. We chose mathematical reasoning datasets MATH (Hendrycks et al., 2021), *competition-level* mathematical problems, and GSM8K (Cobbe et al., 2021), simpler *grade-school level* math word problems. Both datasets are well-established and present a range of difficulty levels. For each dataset, we finetune the base LLM on its respective training set to obtain the initial SFT policy ($\pi_{\mathrm{ref}}$). Throughout the paper, model names refer ones initialized from these SFT checkpoints. We employ *full-parameter finetuning* to leverage the models' full capacity (Biderman et al., 2024).

**Baselines**   Our main baseline is the standard PPO framework (Ouyang et al., 2022; Huang et al., 2024), which VinePPO builds on and improves through better credit assignment. We also compare against PPO variants that forego the credit assignment: RLOO (Ahmadian et al., 2024) and GRPO (Shao et al., 2024). For RL-free alternatives, we include RestEM (Singh et al., 2024), a form of iterative rejection finetuning (Yuan et al., 2023; Anthony et al., 2017), and DPO$^+$ (Pal et al., 2024), a working variant of DPO with strong performance on reasoning. Except VinePPO and standard PPO, all other baselines omit explicit credit assignment by design: i.e. they assign the same weight to all the tokens of a response. All methods use the same SFT checkpoint to ensure fair comparison. For each experiment, we choose the best checkpoint based on a held-out validation set. We compare all methods by accuracy (Pass@1) on test sets, measuring the correctness of final answers.

**Training Details and Hyperparameters**   We adopt a binary task reward $\mathcal{R}$ that evaluates final answer correctness against ground truth, following previous work (Pal et al., 2024; Singh et al., 2024). To ensure fair comparison, all methods consume the same number of episodes during training: for each question, we sample eight episodes and go over the dataset 8 times, yielding 64 episodes per question across all methods. For PPO, we first conduct an extensive hyperparameter search (such as KL penalty coefficient, batch size, minibatch size, GAE $\lambda$, number of epochs per iteration) and rigorously implement all established best practices and well-known techniques (Huang et al., 2024; Ivison et al., 2024) (Refer to Appendix C.2 for the full list). This ensures our evaluation reflects PPO's state-of-the-art configuration and its full potential. VinePPO inherits PPO's *exact hyperparameters* and only modifies the advantage estimation, keeping the rest unchanged. This design allows us to isolate the effect of refined credit assignment. For PPO

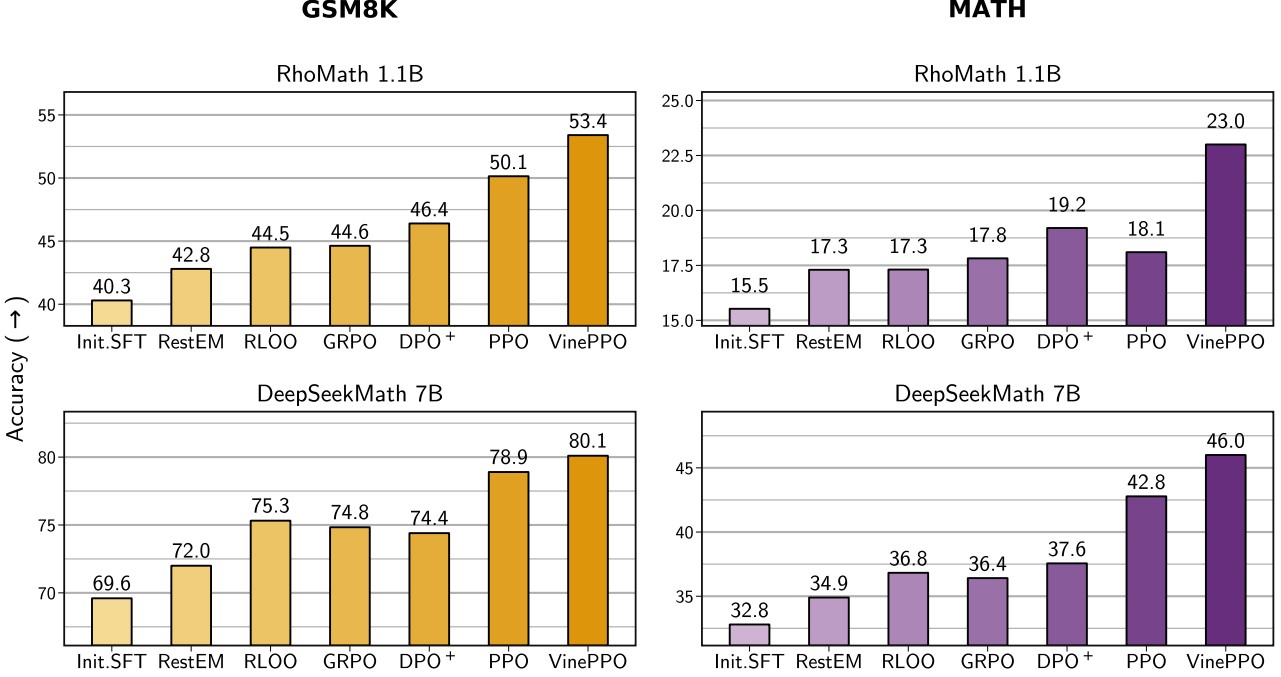

Figure 4: VinePPO outperforms standard PPO, GRPO, RLOO, and other RL-free baselines on Pass@1 performance on MATH and GSM8K datasets, while also exhibiting scalability across different model sizes.

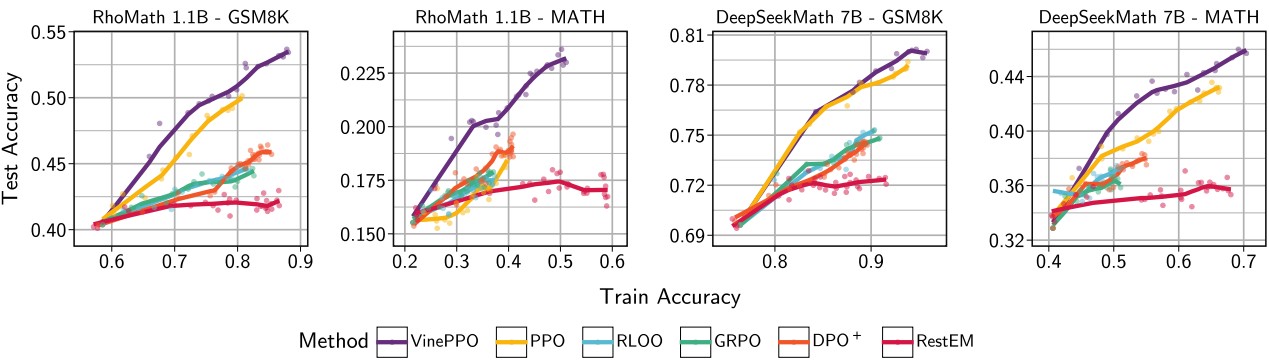

Figure 5: Generalization slope improves with improved credit assignment. VinePPO has steepest generalization: making the highest generalization gains than baselines when fitting the same amount of training data. On the other end of CA spectrum, RestEM overfits its training data.

variants (RLOO, GRPO), we closely follow their Hugging-Face implementations. For these, we initialize with PPO's hyperparameters but perform additional tuning to stabilize training while maintaining the same episode budget. For RL-free baselines (RestEM, DPO+), we strictly adhere to their original implementations (Singh et al., 2024; Pal et al., 2024) and match their sample consumption to other RL methods. For $\hat{V}_{\mathrm{MC}}$ in VinePPO, we conduct a full ablation study on $K$ in Section 6.1, with $K = 9$ used as the default setting unless otherwise specified. To ensure a fair comparison of compute efficiency, we conduct controlled experiments in Section 6.2, where all methods are evaluated under identical

hardware and parallelization protocols. Full implementation details, including hyperparameters and training procedures, are documented in Appendix C.6 to ensure reproducibility.

## 6. Results

In this section, we evaluate the effect of better CA on task performance, efficiency, and generalization dynamics.

### 6.1. Task Performance

VinePPO consistently outperforms standard PPO throughout training (Figure C.4) and other baselines (Figure 4) achiev-

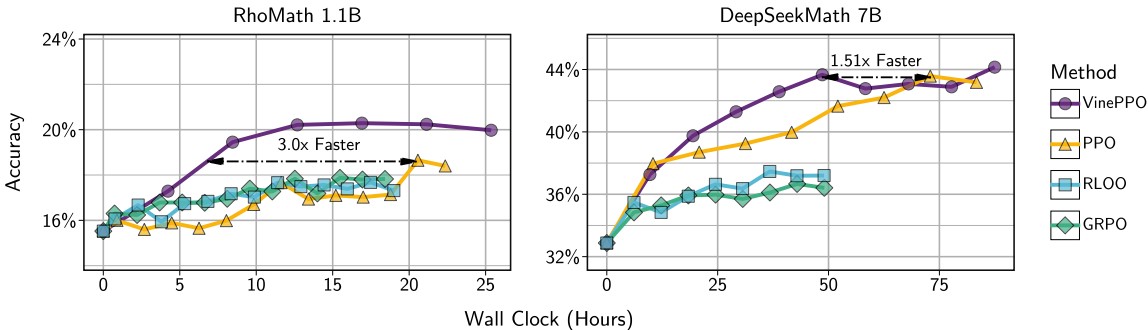

Figure 6: Accuracy vs. Wall Clock Time for both methods measured on the same hardware (shown only up to PPO's final performance). Despite VinePPO taking longer per iteration (up to 2x for 7B and 5x for 1.1B models), it passes PPO's peak performance in fewer iterations and less overall time.

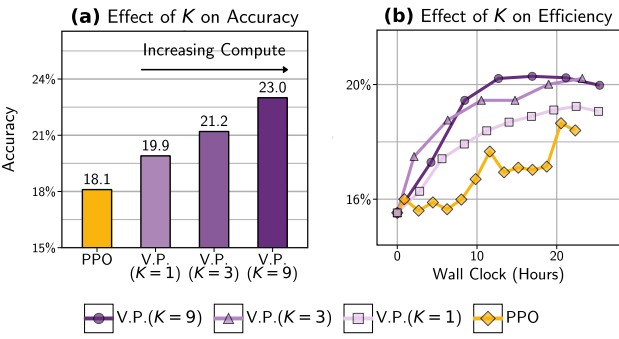

Figure 7: **(a)** Effect of the number of auxiliary rollouts $K$ for estimating $\hat{V}_{\mathrm{MC}}(s_t)$ on RhoMath 1.1B and MATH (see Figure D.11 for GSM8K). Increasing $K$ consistently improves accuracy. **(b)** Wall-clock time for the same experiments. While increased sampling makes each iteration slower, the reduced variance leads to faster overall convergence.

ing the highest test accuracy on both models and datasets. Notably, the performance gap widens in MATH which is more challenging than GSM8K. To confirm that PPO's limitations are not due to undertrained value networks, we measured their *explained variance*, a standard metric for value function quality, which ranged between 0.7–0.9 across tasks (Figure D.5), indicating a well-trained critic. Because the PPO and VinePPO runs only differ in their value estimation, comparing these two isolates the effect of CA. As shown in Figure D.8, VinePPO reaches higher test accuracy given a limited KL budget. Additionally, VinePPO is more robust to higher sampling temperatures (Figure D.10).

## 6.2. Computational Efficiency

Training on a single trajectory in GRPO, RLOO, RestEM and DPO⁺, involves a forward and backward pass. PPO and VinePPO have extra computations of different types. PPO uses double GPU memory — the value network needs

112GB for a 7B LLM, considering both model and its optimizer. Additionally, PPO requires a forward pass for value prediction and a forward-backward pass for value network training. VinePPO replaces the value network with MC samples. Since generation is expensive, each step of VinePPO is slower (up to 5x for RhoMath 1.1B and 2x for DeepSeekMath 7B compared to PPO). VinePPO compensates for slower iterations by making each one more effective through better CA. Under the same hardware, it achieves higher test accuracy faster than baselines (Figure 6). Specifically, VinePPO matches PPO's peak accuracy in *fewer gradient steps* and *less wall-clock time*. Figure 6 shows RhoMath 1.1B and DeepSeekMath 7B require about 3.0x and 1.51x less time and 9x and 2.8x fewer steps compared to PPO. This improvement occurs despite all hyperparameters being tuned for PPO. Therefore, switching to VinePPO could enhance the performance within the same compute budget.

## 6.3. Generalization Slope

High-quality and challenging reasoning tasks are scarce, making generalization a key challenge. Once a training instance is fitted, it provides no further signal for generalization. Thus, algorithms that maximize generalization efficiency are superior—achieving higher test accuracy for a given train accuracy. As shown in Figure 5, VinePPO demonstrates the strongest generalization gains compared to all other baselines. Notably, RestEM overfits near the end. This aligns with recent findings that RL generalizes while SFT primarily memorizes (Chu et al., 2025). Overall, allocating more compute to refining credit assignment, rather than brute-force data fitting, leads to stronger generalization.

## 6.4. Effect of $K$

We assess the impact of $K$, the number of MC samples, by running an ablation on RhoMath 1.1B, varying $K$ from 1 to 3 and 9. As shown in Figure 7, VinePPO improves with

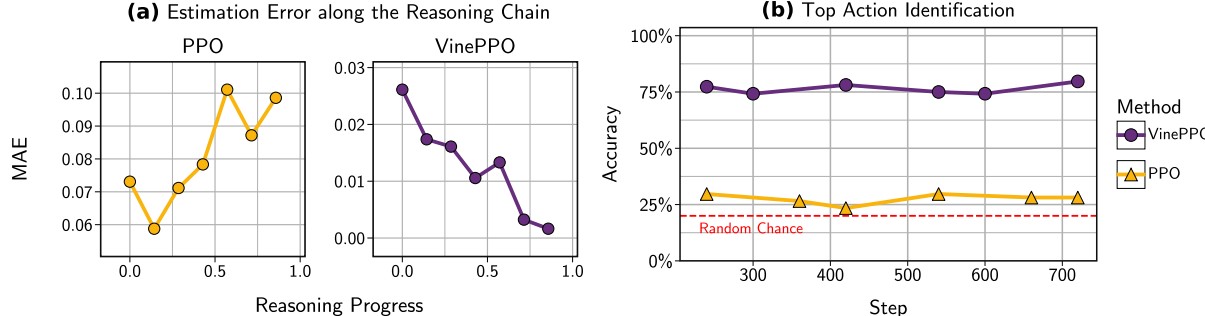

Figure 8: **(a)** Visualizing the Mean Absolute Error (MAE) of the value predictions at different point of the reasoning chain. Value Network in PPO fails to generalize as the reasoning chain progresses, while VinePPO's value estimates become more accurate as the model become more deterministic. **(b)** Accuracy of identifying the top action in a set of five possible next states. VinePPO consistently outperforms the value network.

higher $K$ since more MC samples reduce the variance of $\hat{A}_{\mathrm{MC}}$. While high variance of MC estimation could theoretically hinder training, our results show that even small $K$ values work well in this setting. Interestingly, increasing $K$ also improves compute efficiency. Although each iteration takes longer, it becomes more effective. This suggests that increasing $K$ provides a practical way to leverage additional computational resources for better performance.

## 7. Why and How Value Networks Fail

In this section, we analyze the performance gap between PPO and VinePPO by focusing on their value predictions—their only difference. First, We establish a *"ground truth"* value at each reasoning step within trajectories by running 256 MC samples and averaging the returns. Next, We compare the value predictions against this ground truth[5]. We present the results for DeepSeekMath 7B , our biggest model, on the MATH dataset (all results in Appendix D.5).

**Accuracy**   Figure 3 presents the distribution of value predictions at each reasoning step. VinePPO's estimates are unbiased, with variance peaking at 0.5 and dropping to zero at 0 and 1. PPO's value network shows high bias and often misclassifies bad states (ground truth near 0) as good and vice versa. We define a prediction as "correct" if it is within 0.05 of the ground truth. As shown in Figure D.12 PPO's value network starts with low accuracy, gradually improving to 65%. In contrast, VinePPO consistently achieves 70-90% accuracy throughout training.

**Top Action Identification**   In value-based RL, accurately ranking actions is more important than accurate value esti-

mates. While PPO, a policy-based method, depends heavily on accurate value estimates, it raises an interesting question: Can PPO's value network still rank actions correctly? We tested this by sampling five possible next steps from a shared initial state and measuring whether the method predicted the next step with the highest ground truth value by assigning it the highest predicted value. As shown in Figure 8.b, PPO's value network performs near chance levels for most of the training, improving only slightly over time. In contrast, VinePPO consistently identifies the top action with high accuracy throughout.

**Error Per Reasoning Step**   To understand value estimation dynamics, we plot value estimation error against reasoning step position (normalized; 3rd of 10 steps = 0.3). As shown in Figure 8.a, PPO performs worse as reasoning progresses. We hypothesize this is because early steps resemble training data, allowing the value network to rely on memorization. Later steps are more diverse and value network struggles to generalize. VinePPO's prediction error decreases with reasoning progression. We attribute this to greater determinism in later steps, as the model conditions on a longer context. This stability improves value estimation from the same number of MC samples.

## 8. Discussion

We showed that better credit assignment improves RL training of LLMs. VinePPO is a stepping stone to identify and fix PPO's broken credit assignment. It also opens two future research directions. VinePPO is the first RL post-training algorithm that scales generalization slope by scaling post-training compute. Algorithms that have better generalization trends are valuable given the limited resource of truly challenging and verifiable reasoning tasks. Second, VinePPO highlights the value of reconsidering the implicit assumptions behind default algorithm implementations borrowed

---

[5]The return of sampled trajectory starting from a step is a bernoulli random variable. Let $\theta$ denote the true success probability. Our estimator, $\bar{X} = \frac{1}{256}\sum_{i=1}^{256} X_i$, has variance $\frac{\theta(1-\theta)}{256}$. At $\theta = 0.5$ (maximal variance), this becomes $\frac{0.25}{256} \approx 0.001$.

from Deep RL. In Deep RL, we typically start with a random policy, making it crucial to quickly improve the model's performance. In this context, it's more effective to allocate compute toward gathering additional environment samples rather than perfecting each gradient update. However, with an already capable LLM, it is better to spend more compute to make sure we steer its weights carefully. Overall, we hope VinePPO inspires the community to develop more effective RL training algorithms for LLMs.

## Impact Statement

Our work aims to improve the ability of large language models to perform complex reasoning tasks, potentially contributing to advances in fields such as education, scientific research, and software development. At the same time, more capable reasoning systems could be used irresponsibly, for instance, by automating sophisticated misinformation or other harmful applications. We therefore encourage researchers and practitioners to employ appropriate safeguards when applying our methods. Overall, this paper advances fundamental techniques in machine learning; its societal impact will depend on responsible deployment and continued ethical considerations by the community.

## Acknowledgements

We thank Matheus Pereira for his efforts on facilitating experimentation. AC and NR are supported by CIFAR AI Chair. SR is supported by a Facebook CIFAR AI Chair and NSERC Discovery Grant program. We thank Mila IDT team and Digital Research Alliance of Canada for the compute provided for experimentation.

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

# A. Reviewing PPO

PPO, as used in RL tuning of LLMs, formulates language generation as token-level MDP (Section 3), where each response $\boldsymbol{y}$ is an episode. The state at time step $t$, $s_t \in \mathcal{S}$, is the concatenation of the prompt and the tokens generated so far: $s_t = \boldsymbol{x}; \boldsymbol{y}_{<t} = [x_0, \dots, x_{M-1}, y_0, \dots, y_{t-1}]$. The action $a_t$ corresponds to generating the next token $y_t$ from the model's vocabulary. Given a prompt $\boldsymbol{x}$, an episode of this MDP starts from the initial state $s_0 = \boldsymbol{x}$, and with each action taken, the environment moves to a subsequent state, $s_{t+1} = s_t; [a_t]$, by adding the action $a_t$ to the existing state $s_t$. In the language environment, because states are always formed by concatenating tokens, the environment dynamics are fully known, and the transition function is *deterministic*, meaning $P(s_{t+1}|s_t, a_t) = 1$. Throughout the generation process, the reward $r_t$ is set to zero for all intermediate actions $a_t$, with the sequence-level reward $\mathcal{R}(\boldsymbol{x}; \boldsymbol{y})$ applied only at the final step when the model stops the generation. That is:

$$r_t = r(s_t, a_t) = \begin{cases} \mathcal{R}(\boldsymbol{x}; \boldsymbol{y}) & \text{if } t = T - 1, \text{ where } s_{t+1} = \boldsymbol{y} \text{ is terminal,} \\ 0 & \text{otherwise.} \end{cases} \tag{7}$$

A trajectory $\tau = (s_0, a_0, s_1, a_1, \dots)$ thus represents a sequence of state-action pairs that begins at the input prompt and continues until reaching the terminal state. Finally, the cumulative return of a trajectory $\tau$ is defined as $R(\tau) = \sum_{t=0}^{T-1} r_t = r_{T-1} = \mathcal{R}(\boldsymbol{x}; \boldsymbol{y})$.

The goal of RL tuning is to maximize the expected return of the model's responses to prompts in the dataset, as defined by the reward function $\mathcal{R}$ (Equation 1). PPO, similar to other policy gradient methods, achieves this goal by repeatedly sampling trajectories for a batch of prompt sampled from $\mathcal{D}$ and taking multiple optimization steps in the direction of the gradient $\boldsymbol{g}_{\text{ppo}}$ to update the parameters. PPO gradient $\boldsymbol{g}_{\text{ppo}}$ is defined as the gradient of the following loss:

$$\mathcal{L}_{\text{ppo}}(\theta) = \mathbb{E}_{\tau \sim \pi_{\theta_k}} \left[ \sum_{t=0}^{T-1} \min \left( \frac{\pi_\theta(a_t \mid s_t)}{\pi_{\theta_k}(a_t \mid s_t)} A_t^{\theta_k}, \ \text{clip}(\theta) A_t^{\theta_k} \right) - \beta \, \text{KL}[\pi_\theta \parallel \pi_{\text{ref}}] \right] \tag{8}$$

where $\pi_{\theta_k}$ is the policy at the previous iteration, $\epsilon$ is the clipping parameter, $\beta$ is the KL penalty coefficient, $A_t^{\theta_k} = A^{\theta_k}(s_t, a_t)$ is the advantage estimate for policy $\pi_{\theta_k}$, and the $\text{clip}(\theta)$ function is:

$$\text{clip}(\theta) = \text{clip}\left( \frac{\pi_\theta(a_t \mid s_t)}{\pi_{\theta_k}(a_t \mid s_t)}, 1 - \epsilon, 1 + \epsilon \right). \tag{9}$$

Note that the KL penalty could be also added to the reward function $\mathcal{R}$. We follow the more recent implementations (Shao et al., 2024; Qwen, 2024), where it is added to the loss function. The KL term can be computed using the following unbiased estimator (Schulman, 2020):

$$\hat{\text{KL}}(\theta) = \frac{\pi_{\text{ref}}(a_t \mid s_t)}{\pi_\theta(a_t \mid s_t)} - \log \frac{\pi_{\text{ref}}(a_t \mid s_t)}{\pi_\theta(a_t \mid s_t)} - 1, \tag{10}$$

where $\pi_{\text{ref}}$ denotes the reference model (initial SFT).

## A.1. Value Network

In addition to the policy $\pi_\theta$, PPO also trains a separate value network $\hat{V}_\phi$ to obtain an estimate the true values $V(s_t)$ of states $s_t$. Parameterized by $\phi$, the value network is trained alongside the policy network $\pi_\theta$ using the following loss:

$$\mathcal{L}_{\text{ValNet}}(\phi) = \frac{1}{2} \mathbb{E}_{\tau \sim \pi_\theta} \left[ \frac{1}{T} \sum_{t=0}^{T-1} \max \left( \left\| \hat{V}_\phi(s_t) - G_t \right\|^2, \left\| \text{clip}(\phi) - G_t \right\|^2 \right) \right] \tag{11}$$

where $\hat{V}_{\phi_k}$ is the value network at the previous iteration, $G_t = \sum_{t'=t}^{T-1} \gamma^{t'-t} r_{t'}$ is the empirical return from state $s_t$, $\epsilon'$ is a value clipping parameter, and the $\text{clip}(\theta)$ is defined as:

$$\text{clip}(\phi) = \text{clip}\left( \hat{V}_\phi(s_t), \hat{V}_{\phi_k}(s_t) - \epsilon', \hat{V}_{\phi_k}(s_t) + \epsilon' \right). \tag{12}$$

In RL-tuning of LLMs, the value network is typically initialized from the initial policy $\pi_{\text{ref}}$ (or the reward model, if available), replacing the language modeling head with a scalar output head to predict values (Zheng et al., 2023) This approach takes advantage of the base model's prior knowledge for value estimation.

**Advantage Estimation** Once the estimated values $\hat{V}_\phi(s_t)$ are obtained, the advantages $A(s_t, a_t)$ are computed using the GAE (Schulman et al., 2016):

$$A(s_t, a_t) \approx \hat{A}^{\text{GAE}}(s_t, a_t) \tag{13}$$

$$= (1 - \lambda)\left(\hat{A}_t^{(1)} + \lambda\hat{A}_t^{(2)} + \lambda^2\hat{A}_t^{(3)} + \dots\right) \tag{14}$$

$$= \sum_{l=0}^{\infty}(\gamma\lambda)^l\delta_{t+l} \tag{15}$$

$$= \sum_{l=0}^{\infty}(\gamma\lambda)^l\left(r_{t+l} + \gamma\hat{V}_\phi(s_{t+l+1}) - \hat{V}_\phi(s_{t+l})\right) \tag{16}$$

where $\delta_t = r_t + \gamma\hat{V}_\phi(s_{t+1}) - \hat{V}_\phi(s_t)$ is the temporal difference error, $\lambda$ is the GAE parameter, and $\gamma$ is the discount factor. Also, we have:

$$\hat{A}_t^{(k)} := \sum_{l=0}^{k-1}\gamma^l\delta_{t+l} = r_t + \gamma r_{t+1} + \dots + \gamma^{k-1}r_{t+k-1} + \gamma^k\hat{V}_\phi(s_{t+k}) - \hat{V}_\phi(s_t). \tag{17}$$

Adjusting the GAE parameter $\lambda$ allows for a trade-off between bias and variance in the advantage estimates. However, as we discuss in Appendix C.6, we found that $\lambda = 1$ works best in our experiments (similar to the findings of Trung et al. (2024) and Ahmadian et al. (2024)). In this case, the GAE simplifies to the following form (assuming $\gamma = 1$): $\hat{A}^{\text{GAE}}(s_t, a_t) = \sum_{t'=t}^{T-1} r_{t'} - \hat{V}_\phi(s_t)$.

## B. Reasoning Step Separation Examples

In this section, we outline the methodology used to segment solutions into discrete reasoning steps for the MATH and GSM8K datasets, as illustrated in Figures B.1 and B.2.

For the MATH dataset, we begin by splitting solutions based on clear natural boundaries such as newline characters or punctuation marks (e.g., periods or commas). Care is taken to avoid splitting within mathematical expressions, ensuring that mathematical formulas remain intact. After this initial segmentation, if any resulting step exceeds 100 characters, we further try to divide it by identifying logical breakpoints, such as equal signs (=) within math mode.

For the GSM8K dataset, we take a simpler approach, segmenting the reasoning steps by newlines alone as with this task newlines already serve as natural delimiters.

## C. Experimental Details

### C.1. Datasets

We focus on mathematical reasoning datasets that require step-by-step solutions and are widely used to evaluate the reasoning capabilities of LLMs. Below is a brief overview of the datasets used in our experiments:

**MATH (Hendrycks et al., 2021)** The MATH dataset contains problems from high school math competitions, covering a wide range of topics such as algebra, geometry, and probability. For our experiments, we use the OpenAI split provided by Lightman et al. (2024), which consists of 500 problems for testing and 12,500 problems for training. We further divide the training set into 11,500 problems for training and 500 problems for validation. Each problem includes a step-by-step solution, ending in a final answer marked by \boxed{} in the solution (e.g., "..*so the smallest possible value of c is* $\boxed{\pi}$"). This marking allows for verification of the correctness of model-generated responses by comparing the final answer to the ground truth. We use the scripts provided by Lewkowycz et al. (2022), Lightman et al. (2024), and Shao et al. (2024) to extract and compare the final answers to the ground truth.

**GSM8K (Cobbe et al., 2021)** The GSM8K dataset comprises high-quality grade-school math problems, requiring basic arithmetic or elementary algebra to solve. Although simpler than the MATH dataset, GSM8K is still widely used to assess the reasoning capabilities of LLMs. It contains 1,319 problems for testing and 7,473 for training. To create a validation set, we further split the training set into 7,100 problems for training and 373 for validation. Verifying the correctness of model responses is straightforward, as the final answer is typically an integer, marked by #### in the solution.

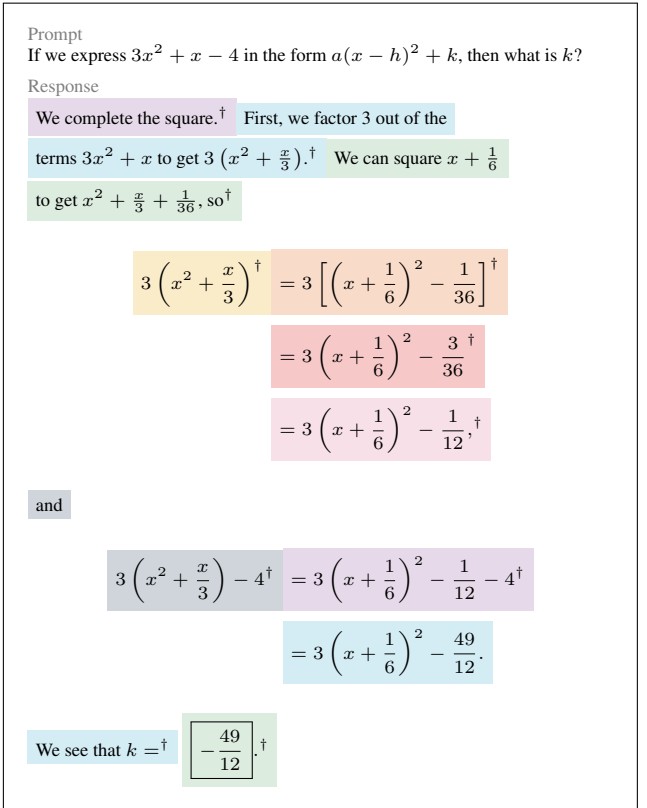

Figure B.1: Examples of solutions separated into its reasoning steps on the MATH dataset. Steps are highlighted using distinct colors. † denotes the reasoning step boundary.

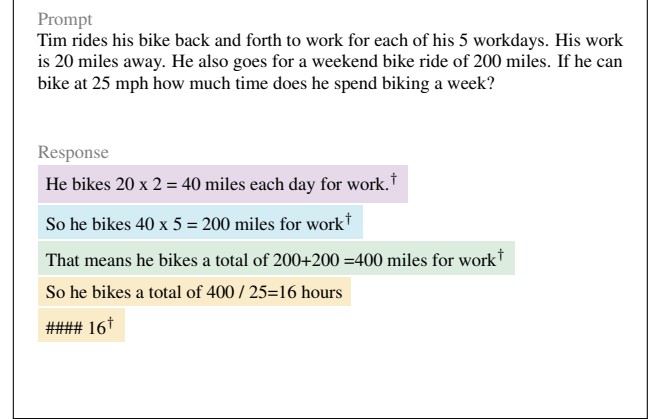

Figure B.2: Examples of solutions separated into its reasoning steps on the GSM8K dataset. Steps are highlighted using distinct colors. † denotes the reasoning step boundary.

### C.2. PPO Implementation

To ensure our PPO implementation is robust, and our evaluation reflects its full potential, we have applied a set of well-established techniques and best practices from the literature (Huang et al., 2024; Ivison et al., 2024; Zheng et al., 2023). Below, we outline the key implementation details that were most effective in our experiments:

- **Advantage Normalization**: After calculating the advantages, we normalize them to have zero mean and unit variance,

Table 1: Summary of PPO hyperparamters used in the experiments.

| Parameter | Value | |
|---|---|---|
| TRAINING | | |
| Optimizer | AdamW | |
| Adam Parameters $(\beta_1, \beta_2)$ | (0.9, 0.999) | |
| Learning rate | $1 \times 10^{-6}$ | |
| Weight Decay | 0.0 | |
| Max Global Gradient Norm for Clipping | 1.0 | |
| Learning Rate Scheduler | Polynomial | |
| Warm Up | 3% of training steps | |
| # Train Steps For MATH dataset | 1000 steps (around 8 dataset epochs) | |
| # Train Steps For GSM8K dataset | 650 steps (around 8 dataset epochs) | |
| GENERAL | | |
| Maximum Response Length | 1024 tokens | |
| Maximum Sequence Length for RhoMath 1.1B | 2048 tokens | |
| Maximum Sequence Length for DeepSeekMath 7B | 2500 tokens | |
| PPO | | |
| # Responses per Prompt | 8 | Search Space: $\{8, 16, 32\}$ |
| # Episodes per PPO Step | 512 | Search Space: $\{256, 512\}$ |
| # Prompts per PPO Step | $512/8 = 64$ | |
| Mini-batch Size | 64 | |
| # Inner epochs per PPO Step | 2 | Search Space: $\{1, 2\}$ |
| Sampling Temperature | 0.6 | Search Space: $\{0.6, 0.8, 1.0\}$ |
| Discount Factor $\gamma$ | 1.0 | |
| GAE Parameter $\lambda$ | 1.0 | Search Space: $[0.95 - 1.0]$ |
| KL Penalty Coefficient $\beta$ | 1e-4 | Search Space: $\{$1e-1, 1e-2, 3e-3, 1e-4$\}$ |
| Policy Clipping Parameter $\epsilon$ | 0.2 | |
| Value Clipping Parameter $\epsilon'$ | 0.2 | |

not only across the batch but also across data parallel ranks. This normalization step is applied consistently in both our PPO and VinePPOimplementations.

- **Reward Normalization**: We follow Ivison et al. (2024) and do not normalize the rewards, as the reward structure in our task is already well-defined within the range of $[0, 1]$. Specifically, correct responses are assigned a reward of 1, while incorrect responses receive 0.

- **End-of-Sequence (EOS) Trick**: As detailed in Appendix A, rewards are only applied at the final token of a response, which corresponds to the EOS token when the response is complete. For responses that exceed the maximum length, we truncate the response to the maximum length and apply the reward to the last token of the truncated sequence. We also experimented with penalizing truncated responses by assigning a negative reward (-1), but this did not lead to performance improvements.

- **Dropout Disabling**: During the RL tuning phase, we disable dropout across all models. This ensures that the log probabilities remain consistent between different forward passes, thereby avoiding stochastic effects that could hurt training stability.

- **Fixed KL Coefficient** We use a constant coefficient for the KL penalty. Although the original PPO implementation for finetining language models (Ziegler et al., 2019) utilized an adaptive KL controller, more recent implementations typically do not use this approach (Ouyang et al., 2022; Touvron et al., 2023; Xu et al., 2024).

## C.3. SFT Models

To ensure a systematic and reproducible evaluation, we create our SFT models $\pi_{\text{ref}}$ by finetuning the *base pretrained LLMs* (as opposed to their "Instruct" version) on the training splits of the respective datasets. Specifically, we produce four distinct SFT models: two base LLM (DeepSeekMath 7B and RhoMath 1.1B ) across two datasets (MATH and GSM8K). The base models are finetuned using the Adam optimizer without weight decay. We employ a learning rate warm-up over 6% of

Table 2: Summary of RLOO and GRPO hyperparamters used in the experiments.

| Parameter | Value | |
|---|---|---|
| TRAINING | | |
| Optimizer | AdamW | |
| Adam Parameters $(\beta_1, \beta_2)$ | (0.9, 0.999) | |
| Learning rate | $1 \times 10^{-6}$ | |
| Weight Decay | 0.0 | |
| Max Global Gradient Norm for Clipping | 1.0 | |
| Learning Rate Scheduler | Polynomial | |
| Warm Up | 3% of training steps | |
| # Train Steps For MATH dataset | 1000 steps (around 8 dataset epochs) | |
| # Train Steps For GSM8K dataset | 650 steps (around 8 dataset epochs) | |
| GENERAL | | |
| Maximum Response Length | 1024 tokens | |
| Maximum Sequence Length for RhoMath 1.1B | 2048 tokens | |
| Maximum Sequence Length for DeepSeekMath 7B | 2500 tokens | |
| RL ALGORITHM | | |
| # Responses per Prompt | 8 | |
| # Episodes per PPO Step | 512 | |
| # Prompts per PPO Step | 512/8 = 64 | |
| Mini-batch Size | 64 | |
| # Inner epochs per PPO Step | 2 | |
| Sampling Temperature | 0.6 | |
| Discount Factor $\gamma$ | 1.0 | |
| KL Penalty Coefficient $\beta$ | 3e-3 | Search Space: {1e-2, 3e-3, 1e-3, 3e-4, 1e-4} |
| Policy Clipping Parameter $\epsilon$ | 0.2 | |

the total training steps. Each model is trained for three epochs with a batch size of 64, and the best checkpoint is selected based on validation accuracy. For each SFT model, we conduct a hyperparameter sweep over learning rates in the range $\{1 \times 10^{-7}, 3 \times 10^{-7}, 1 \times 10^{-6}, 3 \times 10^{-6}, 1 \times 10^{-5}, 3 \times 10^{-5}, 8 \times 10^{-5}, 1 \times 10^{-4}\}$ to ensure optimal performance. We then use these SFT models as the initial checkpoint for training the methods mentioned in our paper.

### C.4. Evaluation

We evaluate each method's performance on the test sets of each dataset. For example, when we report that PPO achieves 42.8% accuracy on the MATH dataset for the DeepSeekMath 7B model, this means the PPO training was initialized with the SFT model specific to DeepSeekMath 7B on the MATH dataset, and accuracy was measured on the MATH test set. Our primary evaluation metric is accuracy, specifically Pass@1, which reflects the percentage of correctly answered problems on the first attempt. This metric is crucial because it represents a realistic user interaction, where the model is expected to deliver a correct answer without the need for multiple tries. For each evaluation, we sample a response from the model for a given prompt, using a maximum token length of 1024 and a temperature of 0.35. A response is considered correct if its final answer matches the ground truth final answer, as detailed in Appendix C.1. Furthermore, each accuracy score is averaged over 16 evaluation rounds, each conducted with different random seeds. This will ensure a robust and low variance assessment of model performance.

### C.5. Other Baselines

**GRPO** (Shao et al., 2024) **and RLOO** (Ahmadian et al., 2024) GRPO replaces PPO's value network with a policy gradient baseline computed from the average return of a group of responses to the same input. For each training question $\boldsymbol{x}$, all algorithms generates $G$ responses, yielding training trajectories $\tau_1, \tau_2, \ldots, \tau_G \sim \pi(\cdot|\boldsymbol{x})$ with corresponding returns $R_1, R_2, \ldots, R_G$. Note that in the case of GRPO, we need to have $G > 1$. Then, GRPO computes the empirical mean $\mu_{\boldsymbol{x}} = \frac{1}{G} \sum_{i=1}^{G} R_i$ and standard deviation $\sigma_{\boldsymbol{x}}$ of these returns. For each trajectory $\tau_i$, the advantage $A(s, a)$ for all state-action

Table 3: Summary of RestEM hyperparamters used in the experiments.

| Parameter | Value | |
|---|---|---|
| | **TRAINING** | |
| Optimizer | AdamW | |
| Adam Parameters $(\beta_1, \beta_2)$ | (0.9, 0.999) | |
| Learning rate | $1 \times 10^{-6}$ | |
| Weight Decay | 0.0 | |
| Max Global Gradient Norm for Clipping | 1.0 | |
| Learning Rate Scheduler | Polynomial | |
| Warm Up | 3% of training steps | |
| | **RESTEM** | |
| # iterations | 10 | |
| # Sampled Responses per Prompt | 8 | Search Space: $\{8, 32\}$ |
| Sampling Temperature | 0.6 | Search Space: $\{0.6, 0.8, 1.0\}$ |
| Checkpoints every # iteration | 500 step | |
| Checkpoint Selection | until validation improves | |
| | Search Space: {until validation improves, best validation} | |

Table 4: Summary of DPO-Positive hyperparameters used in the experiments.

| Parameter | Value | |
|---|---|---|
| | **TRAINING** | |
| Optimizer | AdamW | |
| Adam Parameters $(\beta_1, \beta_2)$ | (0.9, 0.999) | |
| Learning rate | $1 \times 10^{-6}$ | |
| Weight Decay | 0.0 | |
| Max Global Gradient Norm for Clipping | 1.0 | |
| Learning Rate Scheduler | Polynomial | |
| Warm Up | 3% of training steps | |
| | **DPO-POSITIVE** | |
| # DPO-$\beta$ | 0.1 for MATH, 0.3 for GSM8K | |
| # DPO-Positive-$\lambda$ | 50. | |
| # Epochs | 3 | Search Space: $\{3, 8\}$ |
| # Sampled Responses per Prompt | 64 | Search Space: $\{8, 64\}$ |
| # Pairs per prompt | 64 | Search Space: $\{8, 64\}$ |
| Sampling Temperature | 0.6 | |

pairs $(s, a) \in \tau_i$ is defined as:

$$A(s, a) = \frac{R_i - \mu_{\boldsymbol{x}}}{\sigma_{\boldsymbol{x}}}.$$

Notably, this introduces bias in policy gradient estimation because the return $R_i$ of the current trajectory is used in computing its own baseline. RLOO addresses this bias by employing a leave-one-out strategy for baseline computation. Specifically, for each trajectory $\tau_i$, the baseline is computed using the returns of all other trajectories in the group, excluding $R_i$. Let $\mu_{\boldsymbol{x}}^{(i)}$ denote the empirical mean of $\{R_j\}_{j \neq i}$. The advantage for all state-action pairs in $\tau_i$ is then computed as:

$$A(s, a) = R_i - \mu_{\boldsymbol{x}}^{(i)}.$$

This modification ensures that the baseline for each trajectory is independent of its own return, yielding an unbiased policy gradient estimate.

**DPO$^+$ (DPO-Positive) (Pal et al., 2024)** The original DPO method has a failure mode when the edit distance between positive (correct) and negative (incorrect) responses is small. In these cases, the probability of both responses tends to decrease. This issue is especially common in reasoning and mathematical tasks, where multiple solution paths may involve similar equations or steps. Although DPO achieves its goal by reducing the probability of the incorrect response more

than the correct one, it ultimately still lowers the likelihood of generating the correct response. This undermines model performance, making it a failure mode despite partially fulfilling the DPO objective. (Pal et al., 2024; Hwang et al., 2024). While previous methods mitigated this issue by maintaining a high edit distance between positive and negative response pairs, DPO-Positive (Pal et al., 2024) addresses it more effectively. It introduces an additional term to the DPO objective, penalizing any reduction in the probability of the correct response below its probability under the reference model. This ensures that the correct response is not overly suppressed, even when the edit distance is small. The final objective of DPO-Positive is::

$$\mathcal{L}_{\text{DPO-Positive}}(\pi_\theta; \pi_{\text{ref}}) = -\mathbb{E}_{(x,y_w,y_l)\sim\mathcal{D}}\Bigg[ \log\sigma\Bigg( \beta\underbrace{\bigg( \log\frac{\pi_\theta(y_w|x)}{\pi_{\text{ref}}(y_w|x)} - \log\frac{\pi_\theta(y_l|x)}{\pi_{\text{ref}}(y_l|x)} \bigg)}_{\text{DPO Original term}}$$

$$-\lambda\cdot\underbrace{\max\bigg( 0, \log\frac{\pi_{\text{ref}}(y_w|x)}{\pi_\theta(y_w|x)} \bigg)}_{\text{DPO-Positive additional term}} \Bigg) \Bigg] \tag{18}$$

where $\lambda$ is a hyperparameter controlling the weight of the additional term keeping the probabilities of correct responses high. We chose DPO-Positive as a baseline due to its strong performance in (Setlur et al., 2024).

**RestEM (Singh et al., 2024)**  RestEM is an iterative method where, in each iteration, the base model is trained on correct, self-generated responses from the chosen checkpoint of the previous iteration. RestEM takes gradient steps to maximize this objective until the fine-tuned model's accuracy drops on a validation split. The objective of the fine-tuning process is to maximize the log-likelihood of correct responses. Training the model with a maximum likelihood objective on correct responses is mathematically equivalent to training the model with REINFORCE (Sutton et al., 1999), without a baseline, where the entire response is treated as a single action. The reward is 1 when the response is correct, and 0 otherwise. Specifically, we have:

$$\underbrace{\mathbb{E}_{\boldsymbol{x}\sim\mathcal{D},\boldsymbol{y}\sim\pi(\cdot|\boldsymbol{x}),\mathcal{R}(\boldsymbol{x};\boldsymbol{y})=1}\left[\nabla_\theta\log P_\theta(\boldsymbol{y}|\boldsymbol{x})\right]}_{\text{max log-likelihood on correct responses}} = \underbrace{\mathbb{E}_{\boldsymbol{x}\sim\mathcal{D},\boldsymbol{y}\sim\pi(\cdot|\boldsymbol{x})}\left[\nabla_\theta\log P_\theta(\boldsymbol{y}|\boldsymbol{x})\mathcal{R}(\boldsymbol{x};\boldsymbol{y})\right]}_{\text{REINFORCE}} \tag{19}$$

Therefore, maximizing log-likelihood training on correct responses is equivalent to train with policy gradient without precise credit assignment, such as without advantages for specific actions. In our experiments, we observe the impact of this limitation in both Figure C.3 and Figure 5 where RestEM overfits on the training data.

### C.6. Hyperparameters

In this section, we present a comprehensive overview of the hyperparameters used in our experiments. It's important to note that the number of training episodes was carefully selected to ensure that the amount of training data remained consistent across all methods.

**PPO**  Finetuning LLMs using PPO is known to be sensitive to hyperparameter selection, and finding the optimal settings is critical for achieving strong performance. To ensure the robustness of our study, we explored hyperparameter values reported in recent studies (Shao et al., 2024; Zheng et al., 2023; Ivison et al., 2024; Huang et al., 2024) and conducted various sweeps across a wide range of values to identify the best configuration for our tasks and models. Specifically, we find the set of hyperparameters that perform best across both MATH and GSM8K using RhoMath 1.1B model. Then, we employ the optimal set of parameters for the rest of our experiments. The full set of hyperparameters, along with their respective search spaces, is detailed in Table 1.

**VinePPO**  We utilized the same hyperparameter setup as in the PPO implementation (Table 1) for VinePPO.

**RLOO and GRPO**  Since policy optimization in RLOO and GRPO is similar to PPO, we initialze their hyperparameters from PPO. This not only ensure we start from a strong set of values, but also allows for a systematic comparison among these algorithms. We further tune their KL coefficient for stable training. Note that lack of credit assignment mechanism could lead to high variance policy gradient update, resulting in unstable training (Greensmith et al., 2001). See Table 2 for the full list.

Table 5: Average time spent per each training step for different methods and models measured for MATH dataset

| Method | Model | Hardware | Average Training Step Time (s) |
|--------|-------|----------|-------------------------------|
| PPO | RhoMath 1.1B | 4 × Nvidia A100 80GB | 80 |
| VinePPO | RhoMath 1.1B | 4 × Nvidia A100 80GB | 380 |
| PPO | DeepSeekMath 7B | 8 × Nvidia H100 80GB | 312 |
| VinePPO | DeepSeekMath 7B | 8 × Nvidia H100 80GB | 583 |

**RestEM**  To ensure fair comparison we equalize the number of sampled responses for training between our RestEM run and our PPO runs. Therefore, in each RestEM iteration we sample 8 responses per prompt and train for 8 epochs on the correct responses. To enhance RestEM's performance, we also conducted a sweep of other reasonable parameters(Table 3). This included increasing the number of samples to expand the training data and reducing the number of correct responses per question to minimize overfitting.However, we observed no significant improvement .

**DPO$^+$ (DPO-Positive)**  We adopted the same hyperparameters as those used by Setlur et al. (2024). In addition, we conducted a search for the optimal value of $\beta$ to see if using the same $\beta$ as in our PPO experiments would yield better performance than the values they recommended. To maintain a fair comparison, we ensured that the number of training samples in our DPO$^+$ runs matched those in our PPO run where we trained for eight epochs, with each epoch consisting of training on eight responses per question. To match this, we generated 64 pairs of positive and negative responses given 64 self-generated responses from the base model. (Table 4)

### C.7. Compute

All experiments were conducted using multi-GPU training to efficiently handle the computational demands of large-scale models. For the RhoMath 1.1B model, we utilized a node with 4 × Nvidia A100 80GB GPUs to train both PPO and VinePPO. For the larger DeepSeekMath 7B model, we employed a more powerful setup, using a node with 8 × Nvidia H100 80GB GPUs. Additionally, for training DeepSeekMath 7B models with the RestEM approach, we used a node with 4 × Nvidia A100 80GB GPUs. The average training step time for each method on the MATH dataset is presented in Table 5.

### C.8. Software Stack

Both PPO and VinePPOrequire a robust and efficient implementation. For model implementation, we utilize the Huggingface library. Training is carried out using the DeepSpeed distributed training library, which offers efficient multi-GPU support. Specifically, we employ DeepSpeed ZeRO stage 0 (vanilla data parallelism) for RhoMath 1.1B and ZeRO stage 2 (shared optimizer states and gradients across GPUs) for DeepSeekMath 7B . For trajectory sampling during RL training, we rely on the vLLM library (Kwon et al., 2023), which provides optimized inference for LLMs. Additionally, VinePPOleverages vLLM to generate Monte Carlo samples for value estimation. Specifically, after each RL training iteration, the current policy's checkpoint is loaded into vLLM. Then, we use vLLM's serving API to sample new trajectories and also Monte Carlo Samples for VinePPO's value estimation. In our setup, we spawn a separate vLLM engine on each GPU rank. This would allow for data parallelism during both sample generation and training. This software stack ensures that our experiments are both efficient and reproducible. For instance, during VinePPO training, we achieve an inference speed of up to 30K tokens per second using 8 × Nvidia H100 GPUs with the DeepSeekMath 7B model.

### C.9. Reproducibility

In this study, all experiments were conducted using open-source libraries, publicly available datasets, and open-weight LLMs. To ensure full reproducibility, we will release both Singularity and Docker containers, equipped with all dependencies and libraries, enabling our experiments to be run on any machine equipped with NVIDIA GPUs, now or in the future. Additionally, we will make our codebase publicly available on GitHub at `https://github.com/McGill-NLP/VinePPO`

# D. Full Results

## D.1. Training Plots

In this section, we present additional training plots for both PPO and VinePPO on the GSM8K dataset, as shown in Figure D.6. Figure D.7 further illustrates the trade-off between accuracy and KL divergence, while Figure D.9 highlights the computational efficiency of the models[6].

We observe consistent patterns with the results reported in Section 6. Although the performance gap for the DeepSeekMath 7B model is narrower on GSM8K, VinePPO still higher accuracy with significantly lower KL divergence and faster per-iteration time (this happens because responses to GSM8K problems are typically shorter, making MC estimation quite fast).

## D.2. Explained Variance and Mean Absolute Error (MAE) of Value Prediction During Training

To ensure healthy training runs, we assess value prediction accuracy using explained variance and mean absolute error (MAE). Explained variance quantifies how much of the variance in ground-truth values is captured by the estimator:

$$\text{ExplainedVariance} = 1 - \frac{\sum_{g=1}^{n}(v_g - \hat{v}_g)^2}{\sum_{g=1}^{n}(v_g - \bar{v})^2},$$

where $v_g$ are ground-truth values, $\hat{v}_g$ are predictions, and $\bar{v} = \frac{1}{n}\sum_{g=1}^{n} v_g$ is the mean of the ground-truth values. The mean absolute error (MAE) is given by:

$$\text{MAE} = \frac{1}{n}\sum_{g=1}^{n}\left|v_g - \hat{v}_g\right|.$$

As shown in Figure D.5, PPO shows improving explained variance and decreasing MAE, indicating stable training. VinePPO achieves the highest explained variance and lowest MAE. RLOO and GRPO are included solely for demonstration, illustrating the deviation of their baselines from ground truth value estimates.

## D.3. KL Divergence

The RL objective (Equation 1) balances maximizing task performance while constraining deviations from the initial policy $\pi_{\text{ref}}$, measured by KL divergence. We analyze how VinePPO and PPO navigate this trade-off by plotting task accuracy against KL divergence $\text{KL}[\pi_\theta\|\pi_{\text{ref}}]$ throughout training (Figure D.8). Results show VinePPO consistently achieves higher accuracy at same KL divergence, indicating more efficient use of the "KL budget." This efficiency stems from VinePPO's more precise credit assignment. As shown in Figure 1, many advantages are zero, and VinePPO excludes these steps from the loss. By avoiding unnecessary updates on non-contributing tokens, VinePPO reduces non-essential parameter adjustments that would inflate KL.

## D.4. Temperature Tolerance

Sampling temperature is a critical hyperparameter controlling the randomness of sampled trajectories. At higher temperatures models generates more diverse trajectories, accelerating early training through increased exploration. However, this diversity challenges PPO's value network, requiring generalization over a wider range of states. We compared VinePPO and PPO using temperatures $T \in \{0.6, 0.8, 1.0\}$ over the initial third of training steps. Figure D.10 shows VinePPO consistently benefits from higher temperatures, achieving faster convergence. Conversely, PPO fails to benefit from increased exploration and even diverges at $T = 1.0$, where trajectories are most diverse.

## D.5. Value Prediction Analysis

In this section, we provide additional plots for value analysis. Specifically, Figures D.13 to D.16 demonstrates these plots for on the MATH dataset, and Figures D.17 to D.20 on the GSM8K dataset.

Furthermore, we present the prediction error per step in Figures D.21 to D.24.

---

[6]For GSM8K runs of RhoMath 1.1B , different hardware was used, making direct comparison of wall-clock time not feasible.

# E. More Examples of Advantages in VinePPO

In addition to Figure 1, we provide more examples of effective advantage computation of VinePPO in Figures E.25 to E.27.

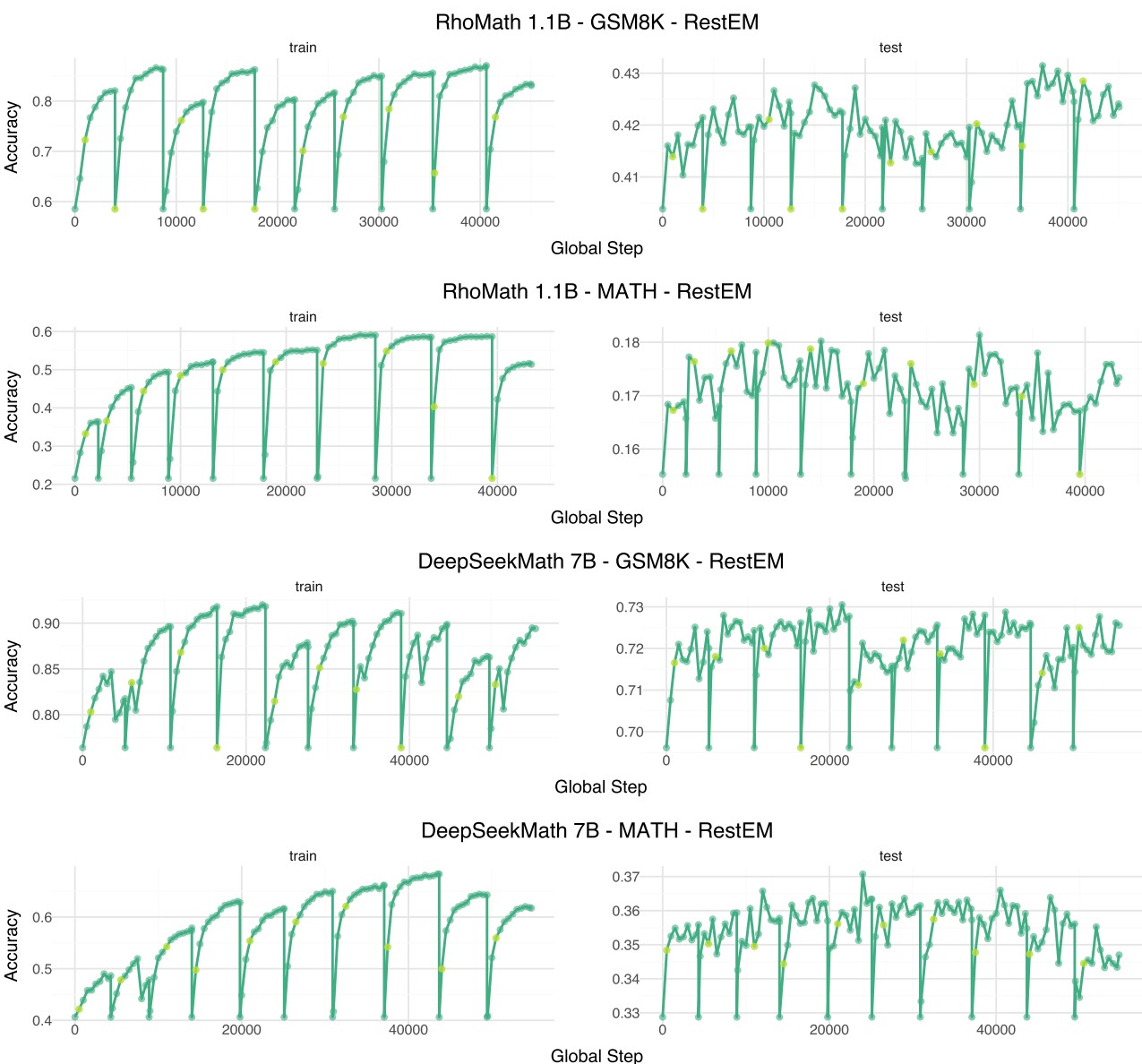

Figure C.3: Performance comparisons across different models and datasets: (a) RhoMath 1.1B on GSM8K, (b) RhoMath 1.1B on MATH, (c) DeepSeekMath 7B on GSM8K, and (d) DeepSeekMath 7B on MATH. The yellow points are chosen checkpoints based on the RestEM rule. Within each iteration, we train on the generated data of the chosen checkpoint for eight epochs and then we choose the first place where performance on a validation split drops following Singh et al. (2024)

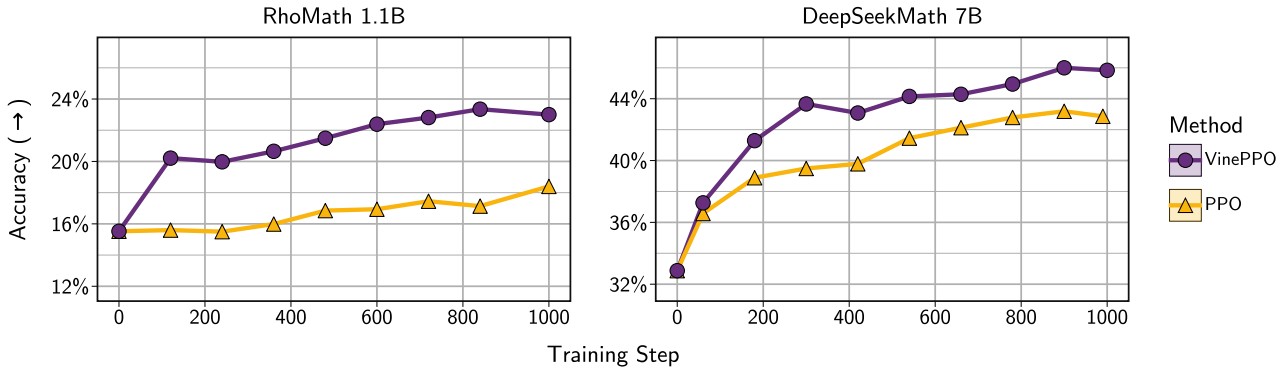

Figure C.4: Comparison of the training behavior between VinePPO and PPO. VinePPO demonstrates consistently higher accuracy (as measured on the test set of MATH dataset) throughout the training. Refer to Appendix D for more detailed plots.

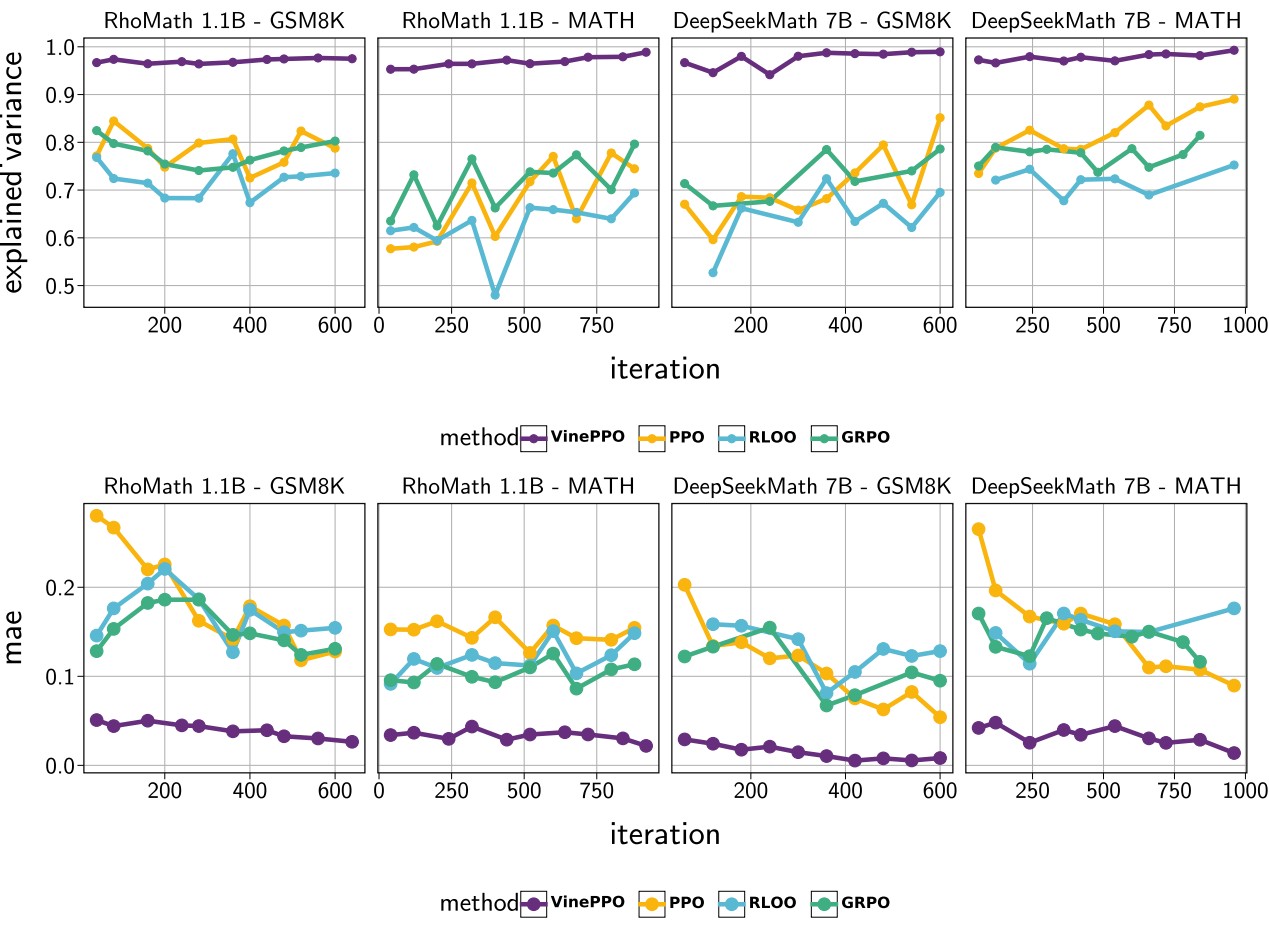

Figure D.5: **Explained Variance and Mean Absolute Error of values**. VinePPO demonstrates higher explained variance in value predictions and lower mean absolute error compared to RLOO, GRPO, and PPO across both datasets. Additionally, PPO's value predictions show non-negative explained variance values close to one, indicating stable and effective training. Note that RLOO and GRPO are included solely for demonstration, illustrating the deviation of their baselines from ground truth value estimates.

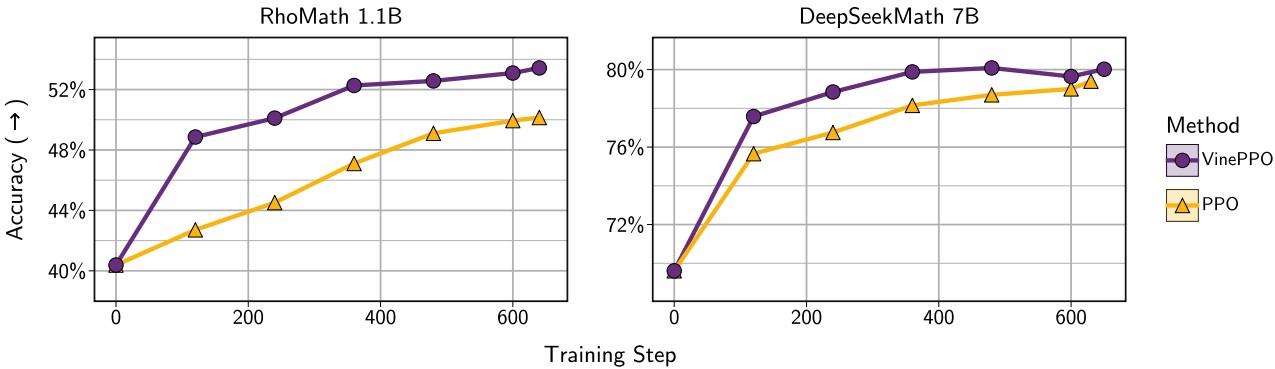

Figure D.6: Comparison of the training behavior between VinePPO and PPO. VinePPO demonstrates consistently higher accuracy throughout the training on the GSM8K dataset. Refer to Figure C.4 for MATH dataset.

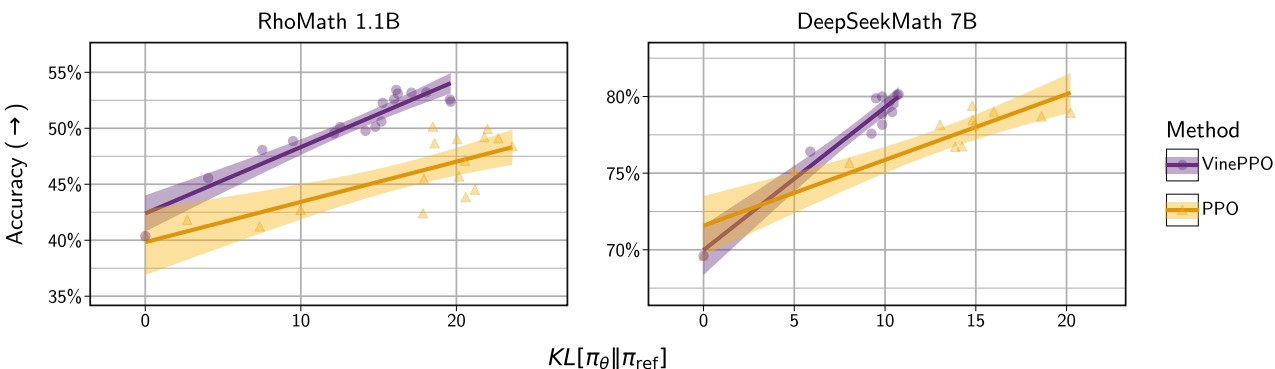

Figure D.7: Task accuracy as a function of KL divergence during training on the GSM8K dataset. VinePPO significantly higher accuracy per KL. Refer to Figure D.8 for MATH dataset.

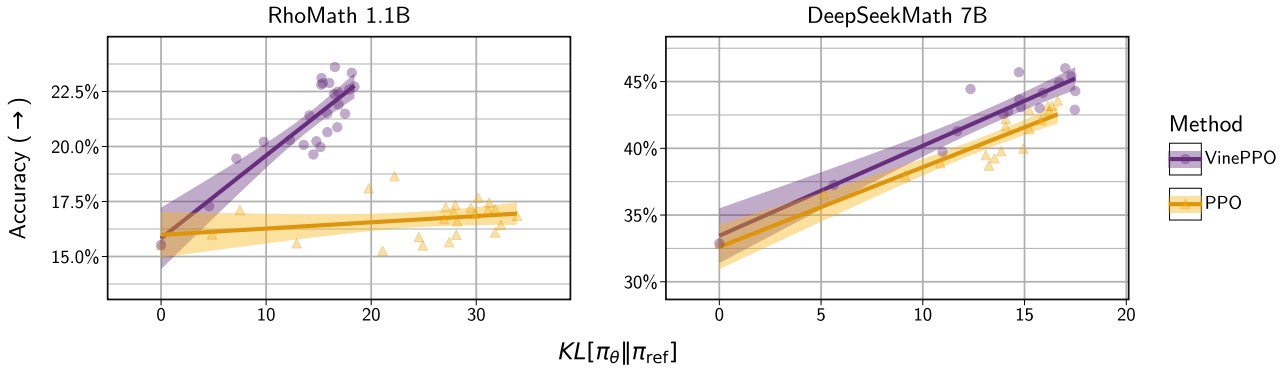

Figure D.8: Task accuracy as a function of KL divergence during training on the MATH dataset. VinePPO achieves higher accuracy, reflecting more efficient credit assignment and focused updates.

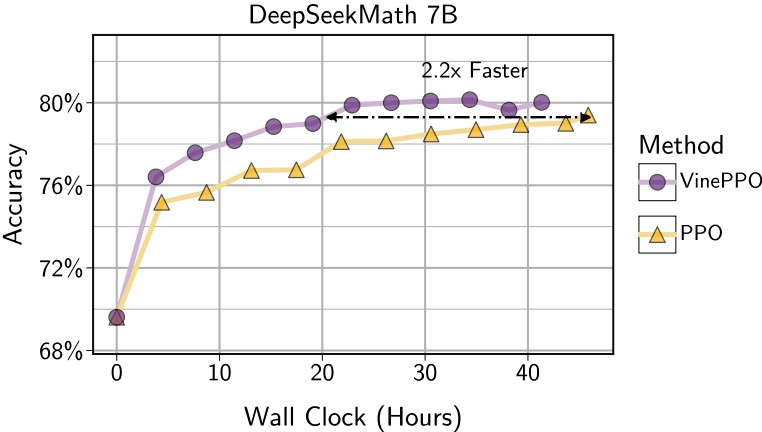

Figure D.9: Accuracy vs. Wall Clock Time for both methods measured on the same hardware throughout the entire training. Since the responses to GSM8K problems are short, VinePPO is even faster per-iteration in our setup and it reaches PPO's peak performance in fewer iterations and less overall time.

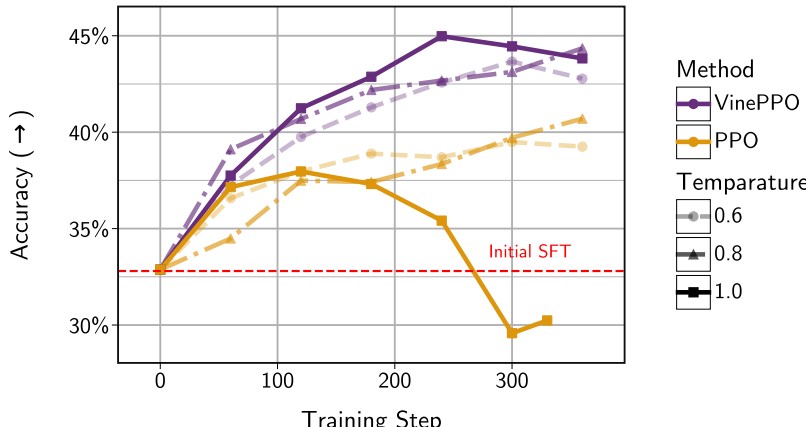

Figure D.10: Test set accuracy during training with higher temperature presented for DeepSeekMath 7B and MATH dataset. VinePPO can tolerate higher temperatures.

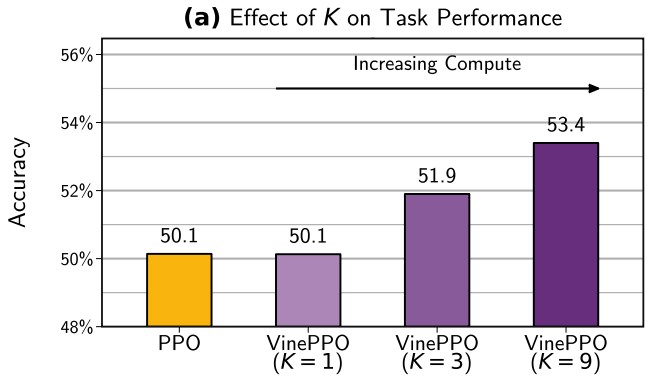

Figure D.11: Ablating the number of auxiliary trajectories $K$ for estimating $\hat{V}_{\mathrm{MC}}(s_t)$ on RhoMath 1.1B and GSM8K. Increasing $K$ consistently improves task performance. (see Figure 7 for MATH dataset)

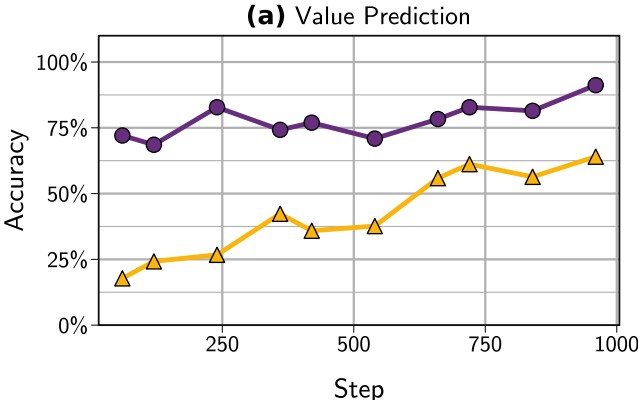

Figure D.12: Value prediction accuracy formulated as a classification problem, where a prediction is considered correct if it falls within 0.05 of the ground truth.

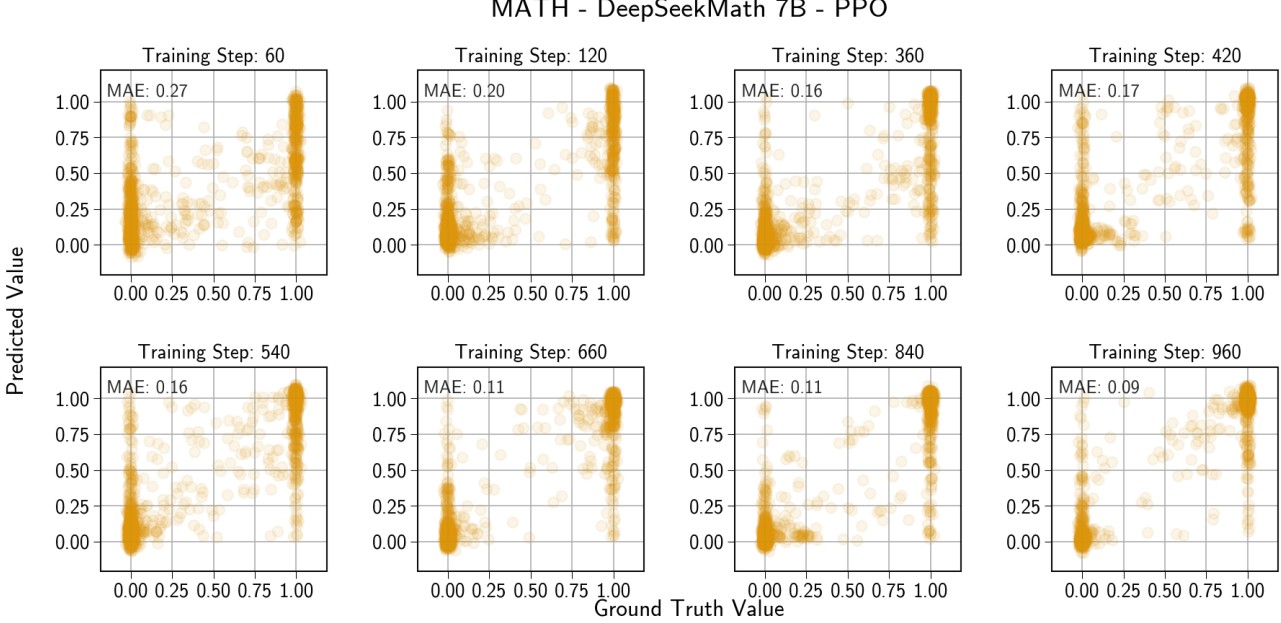

Figure D.13: Distribution of predicted values for each state vs. ground truth (computed using 256 MC samples) during training. MAE denotes the Mean Absolute Error (MAE).

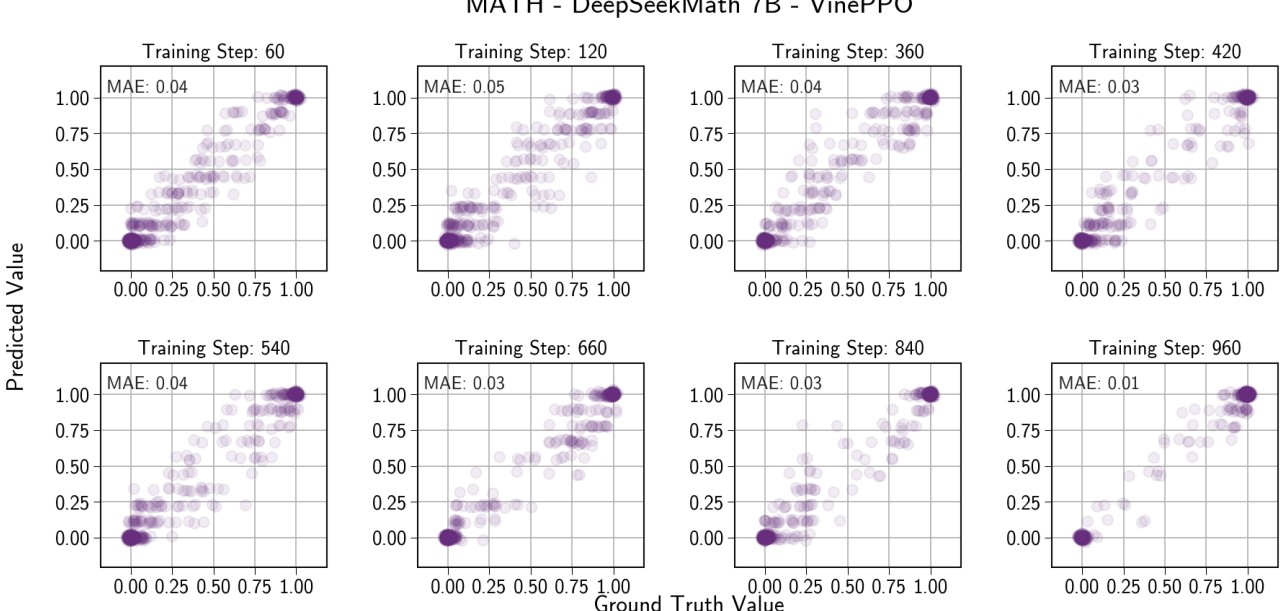

Figure D.14: Distribution of predicted values for each state vs. ground truth (computed using 256 MC samples) during training. MAE denotes the Mean Absolute Error (MAE).

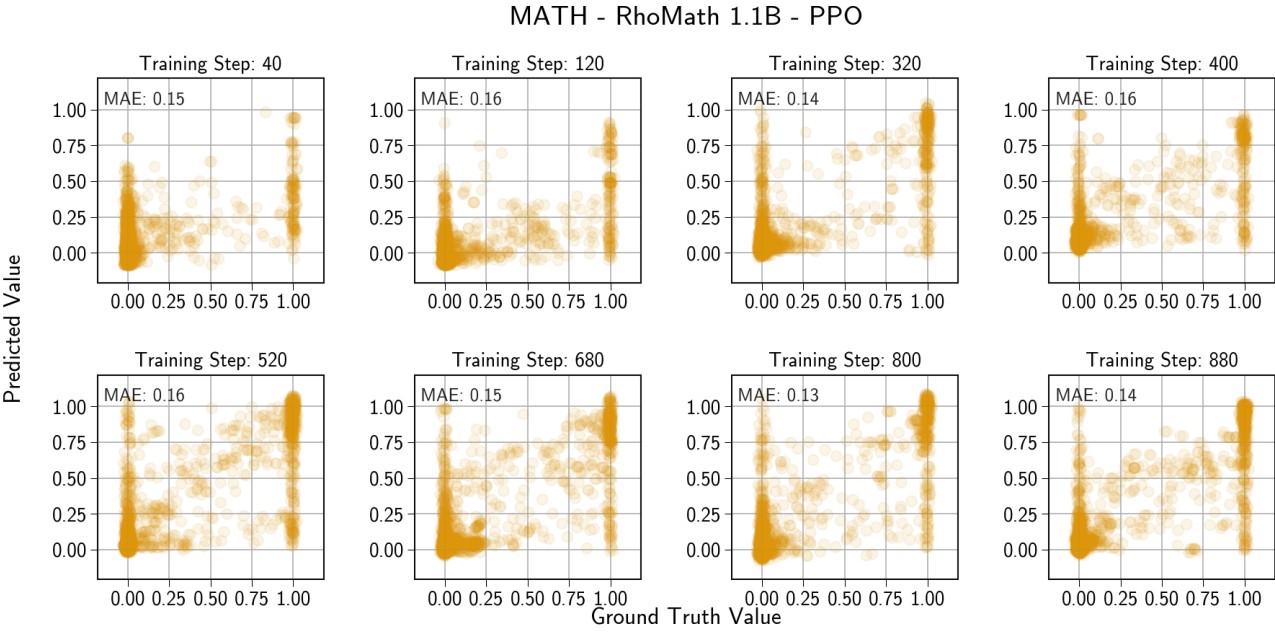

Figure D.15: Distribution of predicted values for each state vs. ground truth (computed using 256 MC samples) during training. MAE denotes the Mean Absolute Error (MAE).

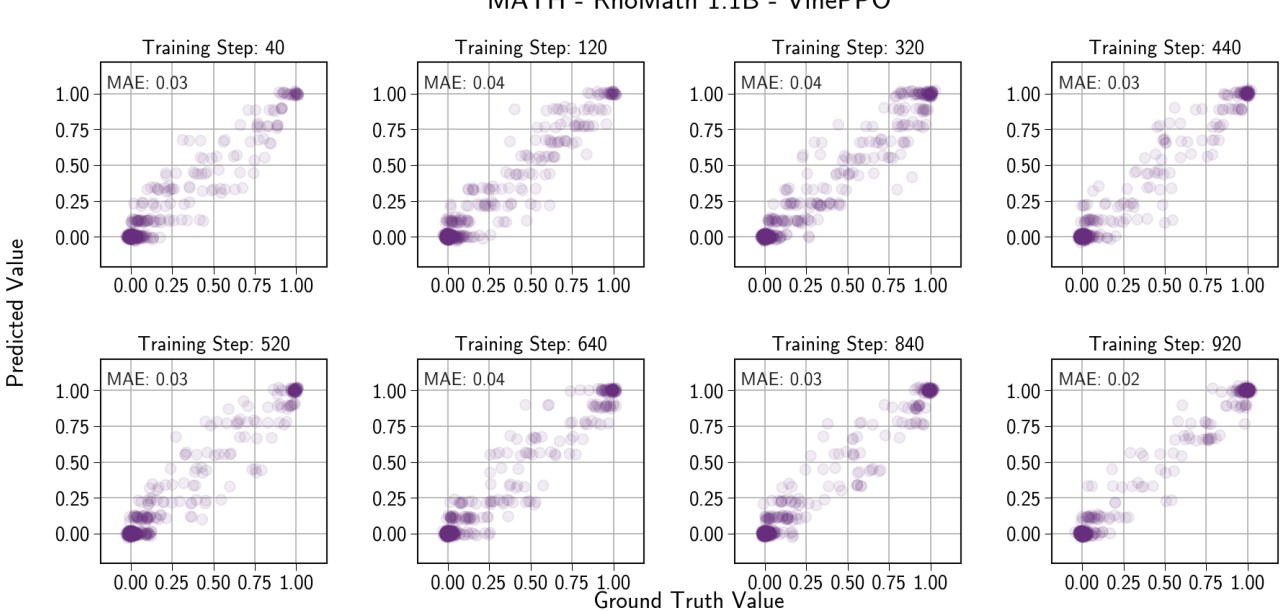

Figure D.16: Distribution of predicted values for each state vs. ground truth (computed using 256 MC samples) during training. MAE denotes the Mean Absolute Error (MAE).

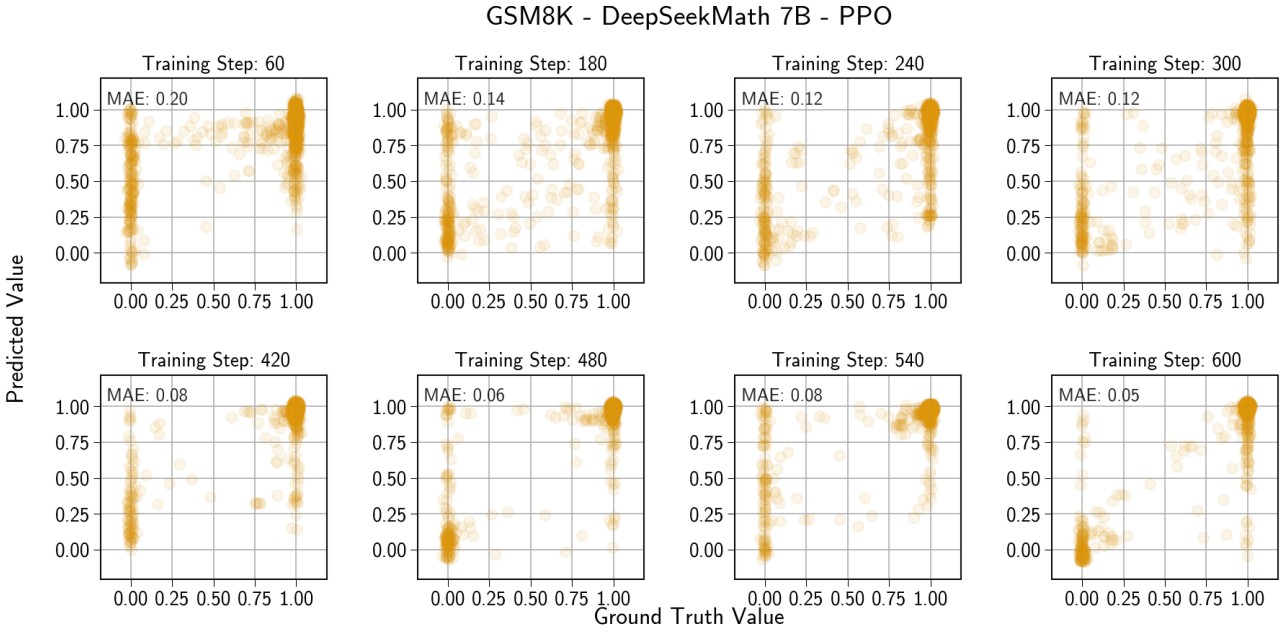

Figure D.17: Distribution of predicted values for each state vs. ground truth (computed using 256 MC samples) during training. MAE denotes the Mean Absolute Error (MAE).

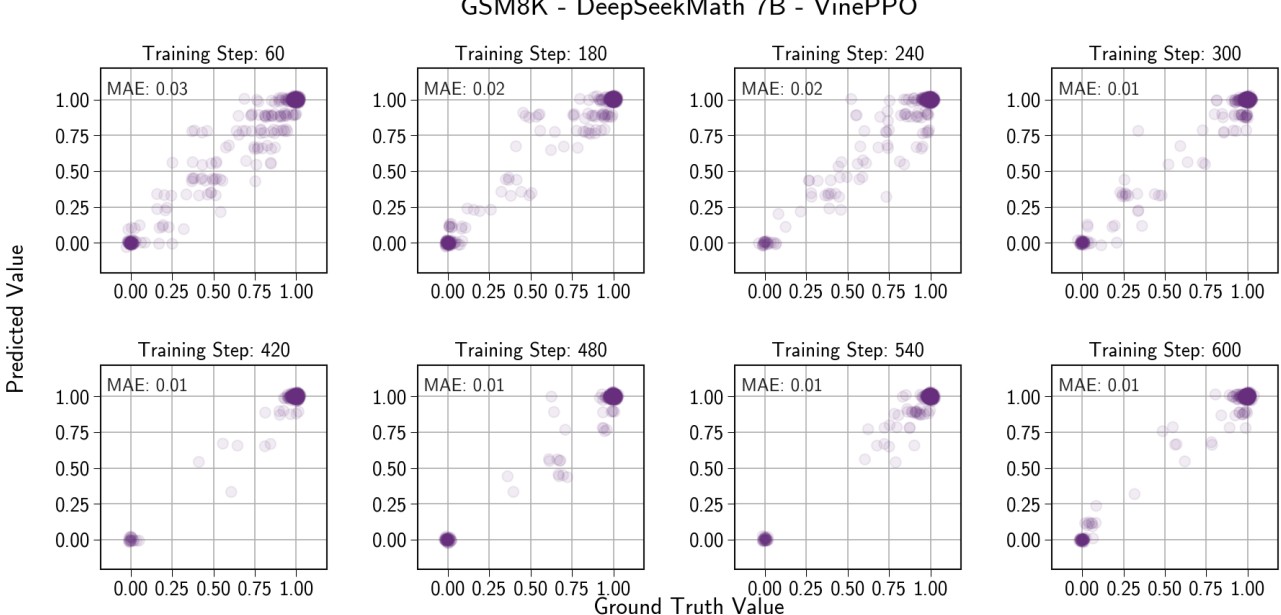

Figure D.18: Distribution of predicted values for each state vs. ground truth (computed using 256 MC samples) during training. MAE denotes the Mean Absolute Error (MAE).

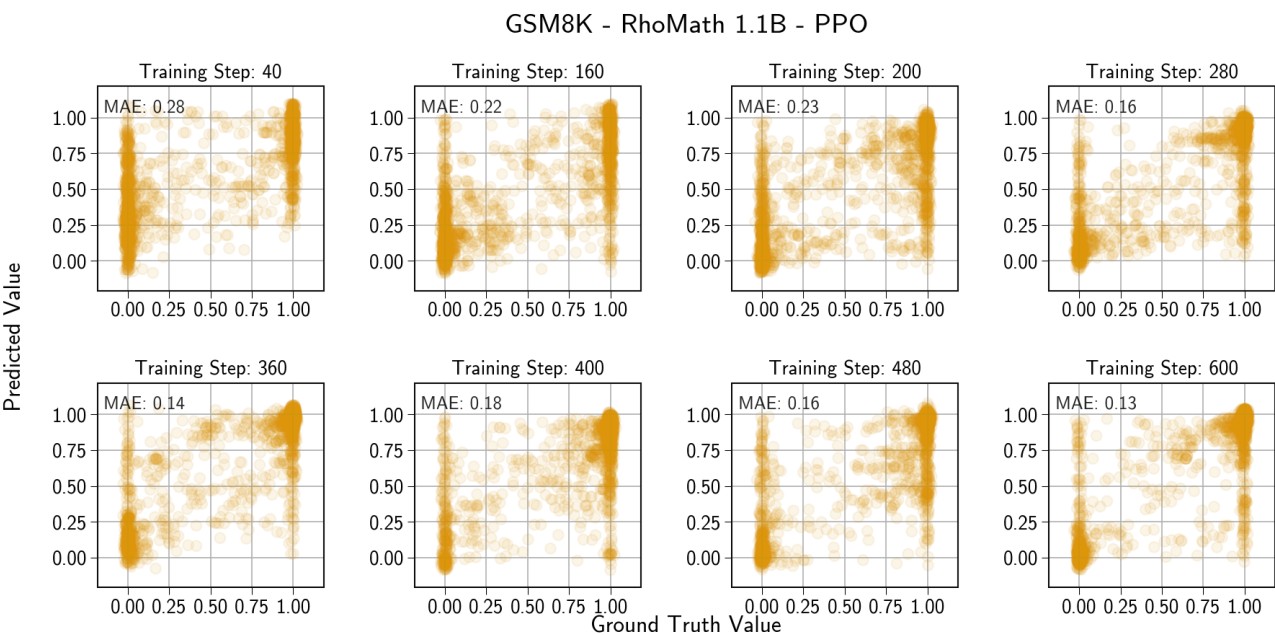

Figure D.19: Distribution of predicted values for each state vs. ground truth (computed using 256 MC samples) during training. MAE denotes the Mean Absolute Error (MAE).

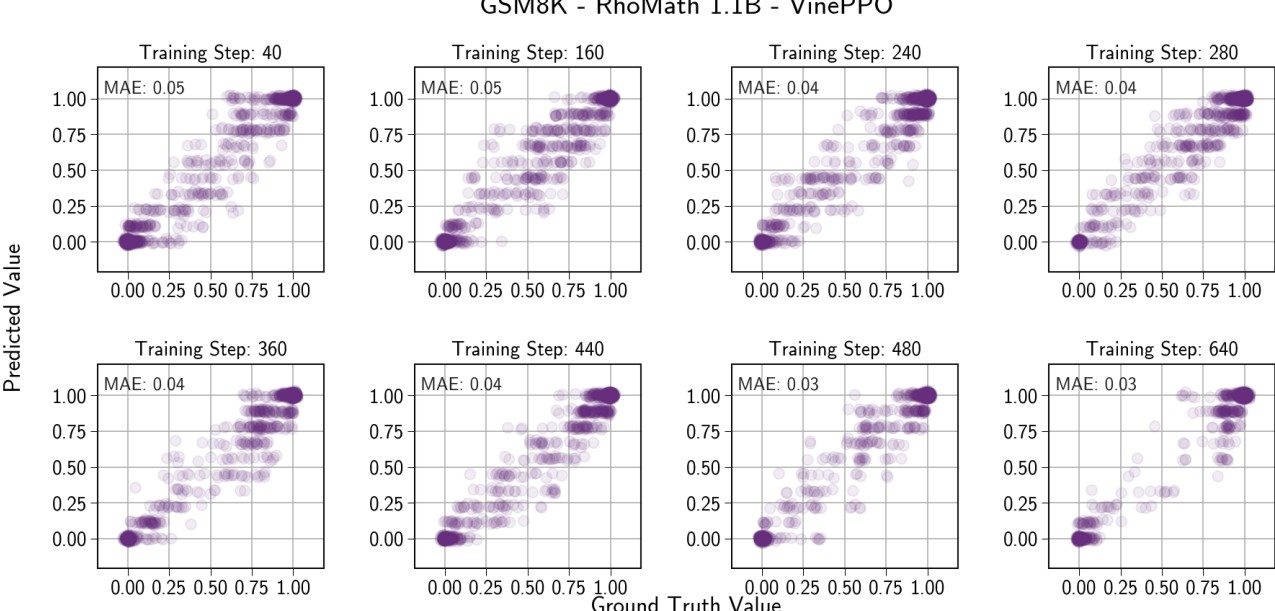

Figure D.20: Distribution of predicted values for each state vs. ground truth (computed using 256 MC samples) during training. MAE denotes the Mean Absolute Error (MAE).

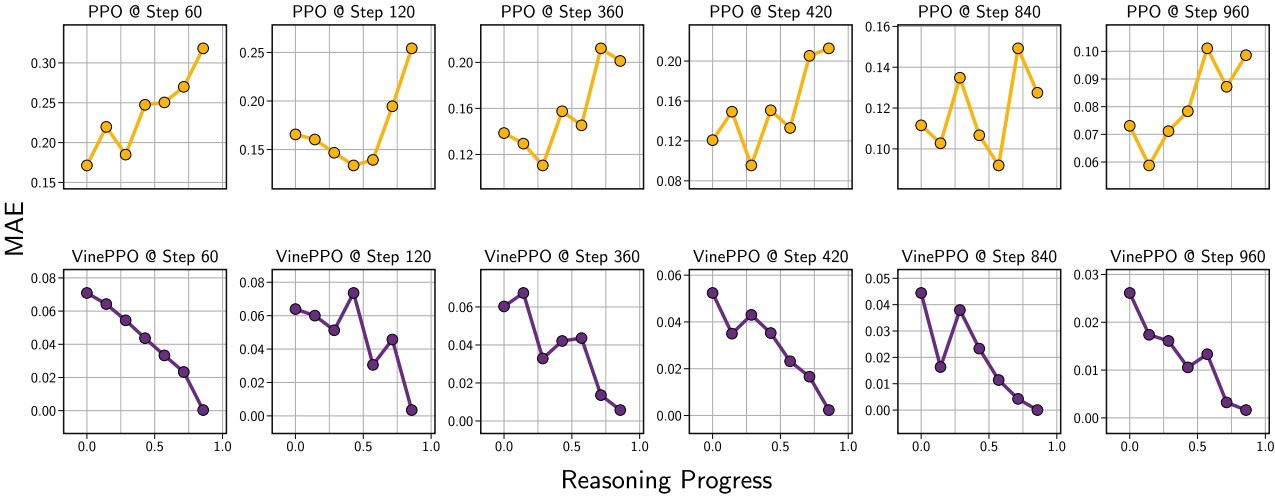

Figure D.21: Visualizing the Mean Absolute Error (MAE) of the value predictions in different point of reasoning chain, plotted for DeepSeekMath 7B on MATH dataset.

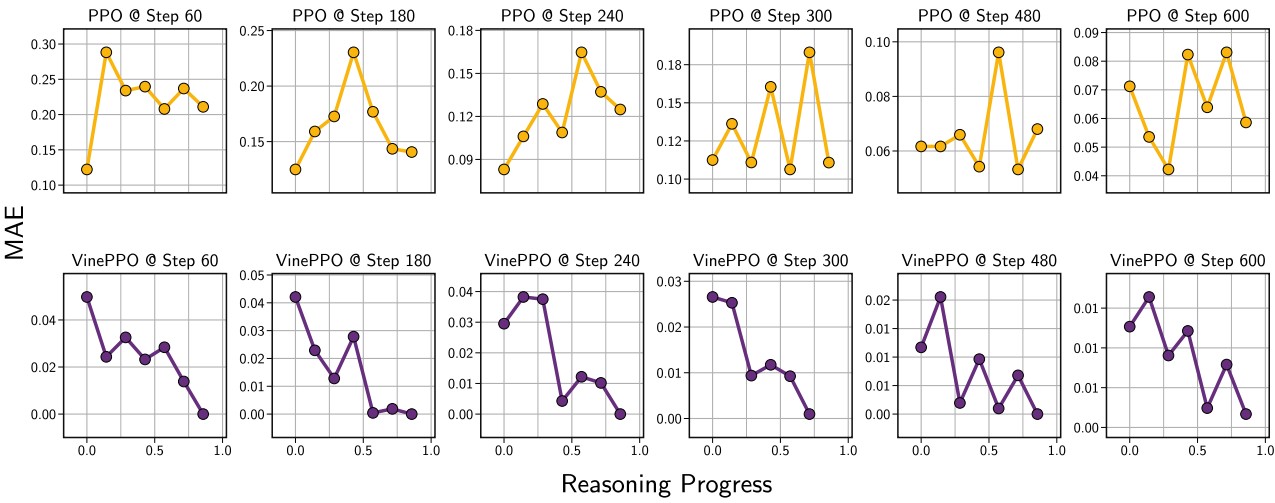

Figure D.22: Visualizing the Mean Absolute Error (MAE) of the value predictions in different point of reasoning chain, plotted for DeepSeekMath 7B on GSM8K dataset.

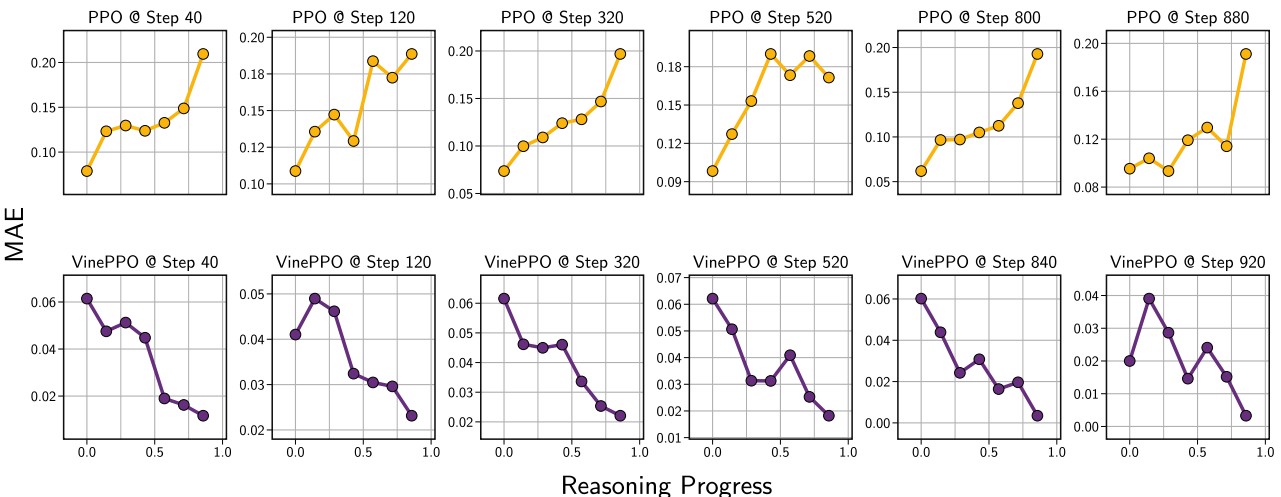

Figure D.23: Visualizing the Mean Absolute Error (MAE) of the value predictions in different point of reasoning chain, plotted for RhoMath 1.1B on MATH dataset.

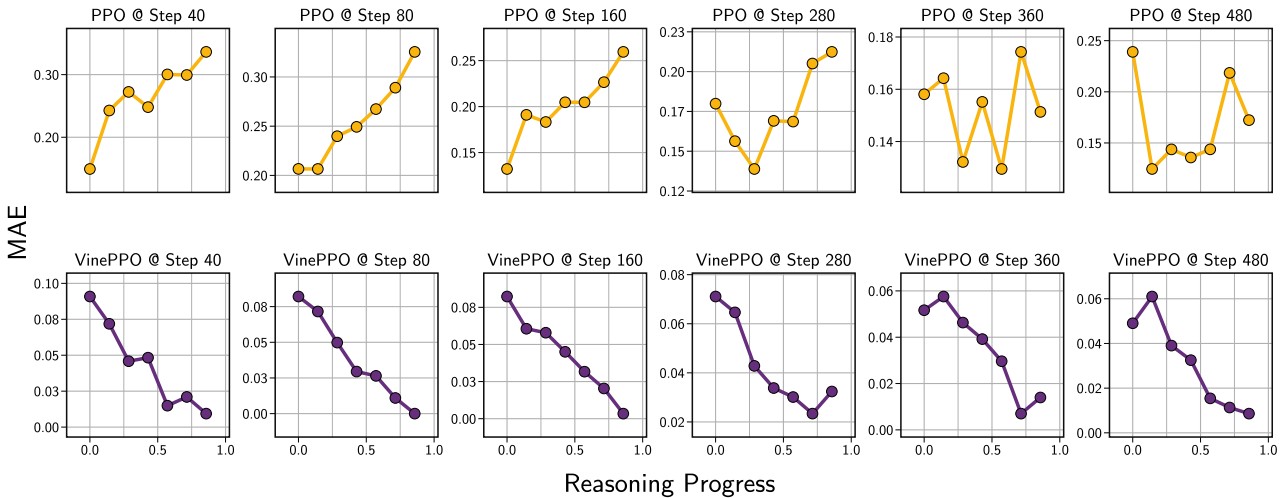

Figure D.24: Visualizing the Mean Absolute Error (MAE) of the value predictions in different point of reasoning chain, plotted for RhoMath 1.1B on GSM8K dataset.

| Prompt ($s_0$) | Advantage |
|---|---|
| The vertex of the parabola described by the equation $y = -3x^2 - 30x - 81$ is $(m, n)$. What is $n$? | |
| **Response** | |
| $s_1$   We can re-write the given equation as $y = -3(x^2 + 10x) - 81$. | 0.0 |
| $s_2$   To complete the square, we add $(-10/2)^2 = 25$ inside the parentheses: | **0.2** |
| $s_3$   $y = -3(x^2 + 10x + 25) - 81 - (-3)(25)$ | **0.5** |
| $s_4$   $= -3(x + 5)^2 - 81 + 75$ | 0.0 |
| $s_5$   $= -3(x + 5)^2 - 6$. | 0.0 |
| $s_6$   The vertex of a parabola in the form $a(x - h)^2 + k$ is at the point $(h, k)$, | 0.0 |
| $s_7$   so the vertex of the given parabola is at the point $(-5, -6)$. | 0.0 |
| $s_8$   Therefore, $n = \boxed{-6}$. | 0.0 |

Example of Advantages Computed by VinePPO

Figure E.25: Steps $s_2$ and $s_3$ provide a critical insight for advancing the solution since they provide a way to organize the equation.

| Prompt ($s_0$) | Advantage |
|---|---|
| Let $f(x) = 2x - 3$ and $g(f(x)) = 5 - 4x$. Find $g(4)$. | 0.4 |
| **Response** | |
| $s_1$  We know that $g(f(x)) = 5 - 4x$. | 0.0 |
| $s_2$  So, if we can figure out what $f(x)$ is when $x = 4$, | **−0.6** |
| $s_3$  we can plug that value into $g(f(x))$ to find $g(4)$. | 0.0 |
| $s_4$  Since $f(x) = 2x - 3$, we have $f(4) = 2(4) - 3 = 5$. | 0.0 |
| $s_5$  Therefore, $g(f(4)) = g(5) = 5 - 4(5) = 5 - 20 = \boxed{-15}$. | 0.0 |

Example of Advantages Computed by VinePPO

Figure E.26: This example demonstrates an error in step $s_2$ where the correct step would be ...*what $x$ is when $f(x) = 4$.* This mistake prevents to model to successfully solve the question after $s_2$.

| Prompt ($s_0$) | Advantage |
|---|---|
| Write the expression $\frac{4+3c}{7} + 2$ as a single fraction. | |
| **Response** | |
| $s_1$  We get a common denominator of 7: | 0.0 |
| $s_2$  $\frac{4+3c}{7} + 2 = \frac{4+3c}{7} + \frac{2 \cdot 7}{7}$ | 0.0 |
| $s_3$  $= \frac{4+3c}{7} + \frac{14}{7}$ | **−0.3** |
| $s_4$  $= \frac{4+3c+14}{7}$ | **0.4** |
| $s_5$  $= \boxed{\dfrac{18 + 3c}{7}}$. | 0.0 |

Example of Advantages Computed by VinePPO

Figure E.27: Steps $s_3$ and $s_4$, despite impacting the probability of solving the question, they are not meaningfully incorrect or insightful.

