# OpenReview forum: "VinePPO: Refining Credit Assignment in RL Training of LLMs"
_ICML.cc/2025/Conference — ICML 2025 poster_

### Official Review · Reviewer_3r7L · 2025-03-09

**Overall Recommendation:** 3

**Summary:**

This paper presents to use MC estimate (a well-established method in classical RL) to improve credit assignment in RL training for large language models (LLMs). The authors highlight the limitations of PPO’s value network, particularly in reasoning-intensive tasks, and propose VinePPO, which replaces traditional GAE estimation with unbiased MC estimation, leading to improved performance in mathematical reasoning benchmarks, including MATH and GSM8K.

**Claims And Evidence:**

Good

**Essential References Not Discussed:**

No

**Experimental Designs Or Analyses:**

I have checked experimental designs or analyses. See comments in Strengths And Weaknesses.

**Methods And Evaluation Criteria:**

Fair. See comments in Strengths And Weaknesses.

**Other Comments Or Suggestions:**

NA

**Other Strengths And Weaknesses:**

Strengths
- The paper is well-written and easy to follow.
- Credit assignment is a crucial challenge, not only in RL but also in reward model training, thus making the motivation of this work strong and relevant.

Weaknesses
- MC estimation is one of the classic methods for value function estimation in RL. This paper did not introduce additional novel contributions. In fact, in PPO, GAE with $\lambda=1$ is also a standard form of MC estimation. Given that VinePPO primarily replaces GAE with its special case, the novelty is limited.
- The effectiveness of MC estimation relies on having a sufficiently large number of samples to mitigate variance. However, in the main experiments, VinePPO only uses k=9 samples. With such a limited sample size, MC could suffer from high variance and instability.
- One way to stabilize MC estimation is to massively increase the number of rollout episodes (potentially in the hundreds). But for LLMs, that’s super expensive, making practical use of VinePPO a great challenge.
- Thus it is intuitive to use more samples (e.g., k=256 in Section 7) to validate the effectiveness of MC estimation. However, it cannot support the effectiveness of VinePPO (k=9) in the main results.

**Questions For Authors:**

- Given that estimating the expectation of $E(R)$ from a single episode is intuitively unreliable, why does VinePPO (K=1) still outperform PPO?

**Relation To Broader Scientific Literature:**

This paper refines credit assignment via MC estimation in RL-based LLM training, contrasting with prior work like PPO, which relies on value networks, and DPO/GRPO, which discard fine-grained credit assignment.

**Theoretical Claims:**

NA

---

> ### Author Rebuttal · Authors · 2025-03-26
>
> We thank the reviewer for their time and effort in reviewing the paper. We now address the concerns raised in the review.
>
> **Novelty**
> > MC estimation is one of the classic methods for value function estimation in RL. This paper did not introduce additional novel contributions. In fact, in PPO, GAE with $\lambda =1$
>  is also a standard form of MC estimation. Given that VinePPO primarily replaces GAE with its special case, the novelty is limited.
>
> *VinePPO is fundamentally different from setting the λ = 1 in GAE within PPO and does not replace GAE.* In fact, both PPO and VinePPO use GAE. Notably, GAE takes value estimates as inputs to compute advantages. The key difference lies in how these value estimates are computed: VinePPO use auxiliary rollouts while PPO trains a critic network. In both methods, GAE’s λ is still a hyperparameter. Our hyperparameter search (see section C.6.) found λ = 1 for PPO and  λ = 0 for VinePPO.
>
> Estimating the values of intermediate states via auxiliary MC was known to be theoretically possible but never deemed practical in DeepRL [1]. This is because auxiliary rollouts require resetting the environment to intermediate states: an uncommon assumption in Deep RL. However, language environments inherently support this capability. VinePPO is, to our knowledge and as noted by Reviewer p3iT, the first to exploit this property, turning an idea once considered impractical in DeepRL into an effective method in RL for LLMs. Moreover, this work goes beyond merely demonstrating better performance; it also provides thorough experiments to explain why.
>
> [1] Schulman, John et al. “Trust Region Policy Optimization.” ArXiv abs/1502.05477 (2015)
>
>
> **Variance and efficiency of MC estimation**
>
> While we agree with the reviewer that in theory variance of MC estimation might necessitate higher values of K, *we empirically establish that even values as small as K=9 demonstrate superior performance (Fig 4), efficiency (Figs 6-7), and stability (Fig D.10) over all baselines*. Our extensive study of compute efficiency in Sec. 6.2 and 6.4 shows that while each iteration takes longer with higher K, the improved credit assignment leads to less wall-clock time for reaching a target test accuracy.
>
> > Thus it is intuitive to use more samples (e.g., k=256 in Section 7) to validate the effectiveness of MC estimation. However, it cannot support the effectiveness of VinePPO (k=9) in the main results.
>
> Please note that in Fig 3, VinePPO values estimates are computed with K=9 and they’re 3x more accurate than the next best baseline in terms of MAE (VinePPO(K=9): 0.03, PPO: 0.11).
>
> **Questions**
> > Given that estimating the expectation of E(R) from a single episode is intuitively unreliable, why does VinePPO (K=1) still outperform PPO?
>
> This is a great question. First, we clarify that K refers to the auxiliary rollouts for value estimation. Regarding the question, both variance and bias can hinder performance. The value network in PPO exhibits high bias in its value estimates (see Fig 3 and Fig D.15). VinePPO even with K=1 provides an unbiased estimate of the value. Although the variance with K=1 might be high, the high bias of the PPO’s value estimates empirically result in worse performance. Note that such subpar performance of value network has been reported in previous work [2].
>
> [2] Ahmadian et al, "Back to Basics: Revisiting REINFORCE Style Optimization for Learning from Human Feedback in LLMs" (2024)
>
> Efficiency was a key focus of our work, and we conducted a thorough analysis of VinePPO’s efficiency in the paper. Across all experiments VinePPO consistently demonstrated superior efficiency and performance compared to baselines. We hope this clarification address your concerns and encourage a fresh evaluation of our work.

---

> > ### Comment · Reviewer_3r7L · 2025-04-03
> >
> > Thanks very much for authors' detailed response. It helped clarify my confusion regarding $\lambda = 1$ and the setup in Section 7. However, I still believe VinePPO has notable limitations.
> >
> > VinePPO requires rolling out K complete trajectories at every token (i.e., at every generation step) to perform MC estimation. This is extremely computationally expensive and resource-intensive. The compute cost increases dramatically with both the response length ($n$) and the MC sample number ($K$).
> >
> > Let’s do a rough calculation (assuming $n = 2048$ and $K = 9$):
> >
> > - Suppose the compute cost for generating each token is normalized to $1$.
> > - For PPO, generating a full response costs $2048$.
> > - For VinePPO:
> >   - Generating the 1st token requires $(2048 – 1) \times 9 + 1$ compute.
> >   - The 2nd token requires $(2048 – 2) \times 9 + 1$.
> >   - The 3rd token: $(2048 – 3) \times 9 + 1$.
> >   - ...
> >   - The 2048th token: $(2048 – 2048) \times 9 + 1$.
> >   - Summing this gives a total cost of $18,867,199$ (based on calculator).
> >
> > This means, for $n = 2048$ and $K = 9$, VinePPO is approximately $9212\times$ more expensive than PPO ($18,867,199\div2048 = 9212$).
> >
> > (Please correct me if there’s any mistake in above calculation.)
> >
> > Given this analysis, the compute overhead appears difficult to justify in practice. However, in Section 6.2, the authors report that VinePPO is only $2\times$ slower than PPO, which seems highly inconsistent with the theoretical estimate above (even if considering responses may terminate before the maximum length).
> >
> > This huge scaling issue with respect to $n$ and $K$ makes it hard to imagine VinePPO working well on recent long reasoning math tasks [1][2], which often involve 8000+ tokens and VinePPO could be tens of thousands of times more expensive. The compute overhead in such settings may quickly become prohibitive.
> >
> > [1] DeepSeek-R1: Incentivizing Reasoning Capability in LLMs via Reinforcement Learning
> >
> > [2] Logic-RL: Unleashing LLM Reasoning with Rule-Based Reinforcement Learning
> >
> > --------
> > If the authors can address the above concerns, I'd be glad to raise my score.
> >
> > ----- update -----
> >
> > **As I am unable to add any more response boxes, I update here**
> >
> > Thank you for the reply. I have increased my score from 1 to 3.

---

> > > ### Author Response · Authors · 2025-04-03
> > >
> > > Thank you for the attention to details and the fruitful discussion started. Your computation respectfully misses fundamental factors:
> > >
> > > **Per-step auxiliary rollouts** You're assuming we perform auxiliary rollouts at every token. We don’t. While per-token rollouts offer the most precise credit assignment (CA), they’re unnecessarily expensive. As explained in Sec 4 (lines 247–252), we divide the response into steps (results in about 10–25 per response in our case) and only apply auxiliary rollouts at step boundaries. This significantly reduces computation: at worst VinePPO's inference is k*(num_steps) slower.
> > >
> > > **Value Network Overhead** Your computation overlooks the cost of training the value network. It’s as large as the base LLM and requires its own optimizer, etc. For a 7B model (& mixed precision training), the value network and optimizer need an extra 112 GB of GPU memory, doubling the total cost. This limits minibatch sizes, and if enough GPU memory is unavailable, optimizer states should be transferred to CPU, significantly increasing communication overhead and training time.
> > >
> > > That’s why in our setup, each **RL step** of VinePPO is only 2× and 5× slower than PPO, for the 7B model and 1B respectively: measured on highly optimized implementations (of both algos) and the exact same hardware: (see our anonymous repo https://anonymous.4open.science/r/icml2025-submission-7155)
> > >
> > > **Effectiveness of Each RL Step** The speed of each RL step doesn’t tell the full story. While VinePPO is slower-per each step, as shown in Fig. C.4, the extra compute yields more effective updates: VinePPO reaches PPO’s peak accuracy in 9× fewer steps on the 1B model and 2.8× fewer on the 7B. We discuss this in sec 6.2.
> > > **As a result, in overall wallclock time, VinePPO is faster than other baseline in reaching any target accuracy.**
> > >
> > > On long reasoning chains: yes, MC rollouts become costlier as trajectories lengthen, but so does the difficulty of credit assignment. Recent work [1] (post submission) shows the effectiveness of similar step-wise MC for long CoT.
> > >
> > > [1] Qu, Yuxiao et al. “Optimizing Test-Time Compute via Meta Reinforcement Fine-Tuning.” (2025).
> > >
> > > ## Broader Picture
> > > While we extensively study the efficiency of VinePPO and show its efficiency superiority in sec 6.2-6.4, figs 6, 7, C.4, D.9, D.11, VinePPO is ultimately about the importance of fine-grained CA in RL for LLMs. CA, despite its importance in DeepRL, has become an overlooked aspect of RL for LLMs, with newer approaches like GRPO, RLOO, DPO removing it entirely.  The broader issue is that ignoring CA harms generalization. See Fig 5: VinePPO outperforms all other baselines in **generalization slope**. In RL for LLMs, high quality verifiable training tasks are limited. Most methods consume them quickly with little gain. VinePPO shows that investing in accurate advantage estimation leads to much better generalization.
> > > *In a nutshell, RL for LLM algorithms should be judged by test accuracy, not faster overfitting on training data.*
> > >
> > > Thanks again for your engagement and we hope this clarification address your concerns and encourage a fresh evaluation of our work.

---

### Official Review · Reviewer_WHfG · 2025-03-12

**Overall Recommendation:** 4

**Summary:**

This paper first analyzes why recent RL methods for LLM often ignore the critic component by showing the poor quality of the learned critic. To improve the critic component, the paper proposes a particularly simple but effective approach by running additional Monte Carlo samples to refine the value estimation, which is called the VinePPO algorithm. The algorithm works strongly on math reasoning benchmarks.

## update after rebuttal
nothing changed. Good paper and I vote for acceptance.

**Claims And Evidence:**

Yes. The paper is very easy to follow with clear statements and solid experiments.

**Essential References Not Discussed:**

The citations are sound.

**Experimental Designs Or Analyses:**

The experiment follows the standard evaluation protocol of math reasoning tasks and RL training. They have also conducted sufficient ablation studies on the algorithmic components.

**Methods And Evaluation Criteria:**

The method is intuitive and effective. In fact, our group has reproduced this method and it really works. The math reasoning benchmark is a popular testbed and the evaluation process in this paper is promising.

**Other Comments Or Suggestions:**

For this draft, I don't have any specific comments. I just have a very separate question for future research that the authors may consider: how would vinePPO perform when training a reasoning model (e.g., o1/R1) with extremely long COT? How can we decide when to perform MC samples when the COT is really long?

**Other Strengths And Weaknesses:**

I like this work. In fact, this work has already made an impact in the RL-for-LLM community.

**Questions For Authors:**

No question. Good work.

**Relation To Broader Scientific Literature:**

This work directly applies to researchers on LLM. Although the method is specifically for RL training, the main point of the work, i.e., using Monte Carlo samples to enhance the value estimates, should be interesting to an even wide audience.

**Theoretical Claims:**

The derivations are all straightforward.

---

> ### Author Rebuttal · Authors · 2025-04-01
>
> We are thrilled that the reviewer liked our work and happy to hear that our results were successfully replicated. We now address the question.
>
> **Questions**
> > How would vinePPO perform when training a reasoning model (e.g., o1/R1) with extremely long COT? How can we decide when to perform MC samples when the COT is really long?
>
> This is a great question. While it is best answered empirically, we believe that credit assignment becomes even more critical in long-CoT models (such as O1/R1). As trajectories grow longer, the distance between the final reward and each individual action increases, making credit assignment more challenging. We hypothesize that in such settings, methods like VinePPO, which enable fine-grained credit assignment, could be more impactful. Moreover, VinePPO could be further optimized by adaptively deciding when to perform additional MC rollouts, for example, based on computed confidence intervals (also known as stopping criteria), which we leave for future work.

---

### Official Review · Reviewer_6B57 · 2025-03-16

**Overall Recommendation:** 3

**Summary:**

The paper highlights a problem with PPO applied to post-training LLMs: an inaccurate value function leads to poor credit assignment. To fix this, the paper proposes to replace the learned value function with an MC estimation. The experiments show that his approach can outperform classical PPO and other relevant baselines for post training LLMs.

**Claims And Evidence:**

The claims around the flaws of PPO in the context of post training LLMs and the advantages of MC estimates appear sufficiently justified by the experiments.

**Essential References Not Discussed:**

none that I'm aware of

**Experimental Designs Or Analyses:**

I've checked all the experiments in the main paper and they appear sound.

**Methods And Evaluation Criteria:**

The proposed method is sensible for checking whether PPO is held back by an inaccurate value function and benchmark datasets are reasonable for comparing LLM RL post-training methods. The experiments could be even stronger, if the models were also compared on AIME.

**Other Comments Or Suggestions:**

the paper is well written

**Other Strengths And Weaknesses:**

strengths:
- the paper highlights an important problem
- the experiments are convincing and the analysis informative

weaknesses:
- the proposed method is not new, albeit it's not commonly seen for solving CA in RL training LLMs

**Questions For Authors:**

1. In figure 3, the plots for GRPO and RLOO, are they depicting the values for the initial state only or for intermediates too?
2. Why does GRPO and RLOO perform so much worse? Is it a lack of hyperparameter tuning or samples per prompt?

**Relation To Broader Scientific Literature:**

the paper highlights a shortcoming of RL post-training LLMs with PPO, which is a common setting in the literature. the method of using monte carlo samples to estimate values is not new, and has been for example proposed for TRPO. the primary contribution of the paper is that it provides many empirical results that demonstrate that learning a value function is hard in this context and that MC estimates are a viable alternative

**Theoretical Claims:**

the paper introduces no new theory

---

> ### Author Rebuttal · Authors · 2025-04-01
>
> We thank the reviewer for their time and their focus on details in their feedback. We now address the raised concerns:
>
> > The proposed method is not new, albeit it's not commonly seen for solving CA in RL training LLMs
>
> While estimating intermediate-states values via auxiliary MC was known to be theoretically possible, it was never deemed practical in DeepRL [1]. This is because auxiliary rollouts require resetting the environment to intermediate states of an already sampled trajectory: a rare assumption in DeepRL. *In fact, even today, none of standard DeepRL frameworks implement or offer any feature resembling the "Vine" approach*. However, reset to intermediate states come naturally in language environments. VinePPO is, to our knowledge and as noted by Reviewer p3iT, the first to exploit this property, turning an idea once considered impractical in DeepRL into an effective method in RL for LLMs. Furthermore, this work goes well beyond demonstrating improved performance and provides thorough analysis to explain the underlying reasons of such improvement.
>
> [1] Schulman, John et al. “Trust Region Policy Optimization.” ArXiv abs/1502.05477 (2015)
>
> **Questions**
> > In figure 3, the plots for GRPO and RLOO, are they depicting the values for the initial state only or for intermediates too?
>
> Figure 3 includes all steps including the intermediate steps to assess the fine-grained credit assignment throughout trajectories.
>
> > Why does GRPO and RLOO perform so much worse? Is it a lack of hyperparameter tuning or samples per prompt?
>
> This is an interesting, yet not surprising result. GRPO/RLOO lack any fine-grained credit assignments. In other words, they assign the same advantage to all the tokens in a response. This is known to be problematic in RL: actions with negative ground-truth advantage may be boosted if their sampled trajectory succeeds, while highly advantageous tokens may be penalized if their trajectory fails.
> All PPO-based variants, such as VinePPO, GRPO, and RLOO inherit PPO’s hyperparameters. However, the lack of fine-grained CA in GRPO/RLOO (see Fig. 3 for their MAE) led to less stable training, requiring even an additional HP tuning compared to VinePPO (see Section C.6). Please note that the same sample per prompt is used for all experiments. In other words, the number of training trajectories, responses, is equal between all algorithms.
>
> We hope we have adequately addressed all of your concerns and would be happy to clarify further if needed. With this in mind, we hope the reviewer increases their rating of our paper.

---

### Official Review · Reviewer_p3iT · 2025-03-20

**Overall Recommendation:** 4

**Summary:**

1. Problem addressed: Credit assignment (CA) in RL training of LLMs, specifically for reasoning-heavy tasks.
2. Method proposed: VinePPO, which replaces value networks with Monte Carlo (MC)-based value estimation for improved CA.
3. Main results:
   1. VinePPO outperforms PPO and other baselines (GRPO, RLOO, RestEM, DPO+) across MATH and GSM8K datasets.
   2. VinePPO achieves better accuracy with fewer iterations and less wall-clock time.
   3. VinePPO shows higher test accuracy for a given training accuracy.

## update after rebuttal

The rebuttal provides few new things to resolve my concern but I can understand the authors' explanation. So I would keep my score as 4.

**Claims And Evidence:**

The claims made in the submission are supported by clear and convincing evidence.

The only minor issue is that they haven't been examined on larger-scale models over 7B parameters. But I fully understand that this is reasonable considering the resource constraints.

**Essential References Not Discussed:**

Most related works are discussed in the paper.

**Experimental Designs Or Analyses:**

The experimental design is sound and the analyses are valid.

**Methods And Evaluation Criteria:**

The proposed methods and evaluation criteria make sense for the problem or application at hand.

**Other Comments Or Suggestions:**

I have no other comments or suggestions.

**Other Strengths And Weaknesses:**

Strengths:

1. The paper is insightful to leverage the reverting flexibility of the language environment and paralleled efficiency of modern inference engines.
2. The paper obtains significant improvements over existing methods.
3. The paper is well-written and easy to follow.

Weaknesses: No significant weaknesses.

**Questions For Authors:**

1. What about evaluation on more OOD benchmark datasets like AIME?
2. Is VinePPO still effective for larger-scale models?

**Relation To Broader Scientific Literature:**

The paper is well-grounded in the broader scientific literature for the credit assignment problem in RL training of LLMs.

For novelty, previous works

1. either focus on biased, low-variance credit assignment based on the value network, e.g., PPO,
2. or focus on unbiased, high-variance credit assignment, e.g., GRPO, RLOO, RestEM, DPO+,

VinePPO takes the advantage of unbiasedness of Monte Carlo sampling and resolves the problem of high variance by increasing the sampling number, leveraging the reverting flexibility of the language environment.

For performance, VinePPO outperforms PPO and other baselines (GRPO, RLOO, RestEM, DPO+) across the commonly used MATH and GSM8K datasets.

**Theoretical Claims:**

There is no proof nor theoretical analysis in the paper.

---

> ### Author Rebuttal · Authors · 2025-04-01
>
> We appreciate the reviewer’s attention to detail and positive feedback. Below, we address the questions asked:
>
> **Questions**
> > What about evaluation on more OOD benchmark datasets like AIME
>
> We considered benchmarks like AIME too difficult for the base LLMs used in our experiments. However, this is an interesting question that could be explored with newer and stronger models.
>
> > Is VinePPO still effective for larger-scale models?
>
> While this question is ultimately empirical (and unfortunately beyond our current resources), we hypothesize that in larger models, which often produce longer responses, the credit assignment problem could be even more critical. In very long chains-of-thoughts, where reward is only received at the end, attributing credit to individual actions becomes increasingly challenging. Therefore, we believe methods like VinePPO, which offer better credit assignment, should be more relevant in such settings.

---

### Decision · Program_Chairs · 2025-05-01

**Decision:**

Accept (poster)

**Comment:**

This paper introduces VinePPO, a method that enhances credit assignment in RL training of LLMs via Monte Carlo-based value estimation. It is a well-executed empirical study that addresses a key challenge in RLHF with strong results on MATH and GSM8K. I recommend acceptance for publication on ICML 2025.